# Can Knowledge-Graph-based Retrieval Augmented Generation Really Retrieve What You Need?

**Junchi Yu**[1], **Yujie Liu**[2], **Jindong Gu**[1], **Philip Torr**[1], **Dongzhan Zhou**[2]*

[1]Department of Engineering Science, University of Oxford, UK
[2]Shanghai Artificial Intelligence Laboratory, China
junchi.yu@eng.ox.ac.uk, liuyujie1@pjlab.org.cn, jindong.gu@eng.ox.ac.uk,
philip.torr@eng.ox.ac.uk, zhoudongzhan@pjlab.org.cn

## Abstract

Retrieval-Augmented Generation (RAG) based on knowledge graphs (KGs) enhances large language models (LLMs) with structural and textual external knowledge. Yet, existing KG-based RAG methods struggle to retrieve accurate and diverse information when handling complex queries. By modeling KG-based retrieval as a multi-step decision process, Process Reward Models (PRMs) offer a promising solution to align the retrieval behavior with the query-specific knowledge requirements. However, PRMs heavily rely on process-level supervision signals that are expensive and hard to obtain on KGs. To address this challenge, we propose **GraphFlow**, a framework that efficiently retrieves *accurate and diverse* knowledge required for complex queries from text-rich KGs. GraphFlow employs a detailed balance objective with local exploration to jointly optimize a retrieval policy and a flow estimator. The flow estimator factorizes the outcome reward of the retrieval results into the intermediate retrieval steps. Such reward factorization guides the retrieval policy to retrieve candidates from KGs in proportion to their outcome reward. This allows GraphFlow to explore relevant regions of KGs that yield diverse and accurate results. We evaluate GraphFlow on STaRK benchmark, which includes real-world queries from multiple domains over text-rich KGs. GraphFlow outperforms strong KG-based RAG baselines including GPT-4o by 10% performance gain on both retrieval accuracy and diversity metrics. GraphFlow also shows strong generalization by effectively retrieving information from unseen KGs to support new-domain queries, highlighting its effectiveness and robustness [2].

## 1 Introduction

Retrieval-Augmented Generation (RAG) [36] has emerged as a promising framework to reduce the hallucination of Large Language Models (LLMs) by mitigating the gap between model knowledge and factual knowledge [28, 89, 26]. Traditional RAG usually employs an unstructured vector-indexed database as the external knowledge source, where the text corpus is indexed using pretrained encoders to support retrieval. Recent work has explored knowledge graphs (KGs) [68] as a structural alternative to the external knowledge source of RAG [59]. KGs enjoy several advantages over the vector-indexed database in traditional RAG, such as representing relational information with graph structures [88], integrating knowledge from heterogeneous resources [59], and enhancing interpretability by neural-symbolic reasoning [87]. Thus, KG-based RAG has demonstrated great potential in enhancing LLMs in many domains, including medical diagnosis [67], biochemistry [77], and physics [72].

---

*Corresponding Author

[2]Code is available https://github.com/Samyu0304/GraphFlow

Recent KG-based RAG methods employ two approaches to retrieve information from KGs when receiving an input query [47, 19, 48]. Retrieval-based approaches [21, 37, 80] leverage pretrained language models [62] to encode KG text into embeddings, and use retriever models [31] to identify relational triplets or subgraphs in KGs that most support the query. And agent-based methods treat LLMs as searching agents to navigate across KGs and retrieve a relational path with supporting information for a given query [68, 47, 50].

While KG-based RAG methods show promise in retrieving structural information for simple relational queries, their effectiveness is limited in more complex ones. As shown in Figure 1, structural information in KGs alone is often sufficient for many relational queries. For example, retrieving the relation triplet $(Alice, daughter, Bob)$ adequately answers the question "Who is the father of Alice?".

However, complex queries typically require leveraging both structural and textual information during retrieval [50]. Consider the paper searching query "Please list the papers published by University A relevant to research topic B". Addressing such a query requires understanding authorship and affiliation relationships, as well as text descriptions of paper content, research topics, institutions, and authors.

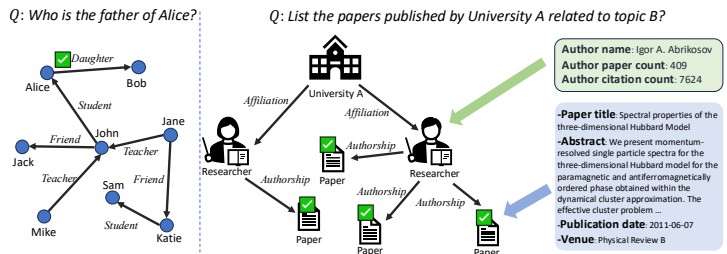

Figure 1: Comparison of the retrieval tasks between relational and complex queries.

The fusion of relational and text knowledge presents a significant challenge for accurate retrieval with KG-based RAG methods. Another challenge is the diversity of retrieval targets in complex queries. Unlike relational queries corresponding to a single deterministic retrieval target (e.g., $Alice \xrightarrow{daughter} Bob$), complex queries require retrieving a diverse set of candidates. For example, the paper searching query in Figure 1 corresponds to multiple retrieval targets. Therefore, KG-based RAG must emphasize both retrieval accuracy and diversity when supporting complex queries. Unfortunately, our empirical findings reveal that existing KG-based RAG methods face challenges in achieving this goal.

To overcome the above challenge, it is essential to align the retrieval process of KG-based RAG with the diversity and accuracy demands for complex queries. Process Reward Models (PRMs) [54, 90, 92, 85] offer a promising framework for this goal. By providing step-wise guidance, PRMs have been widely used in LLM alignment [40], reasoning [7] and planning [6] when treating these tasks as the multi-step decision. In KG-based RAG, the retrieval process can be naturally viewed as a multi-step decision process, where an agent traverses a KG and expands its retrieval trajectory at each step. PRMs can provide step-wise guidance for the agent to retrieve desired information for complex queries. However, training a PRM needs high-quality preference datasets with fine-grained and process-level reward signals [57, 76, 27]. In KG-based RAG, assessing the process-level reward at each step in retrieval trajectories is expensive. Only the outcome reward is easily available (i.e., whether the retrieval trajectory can support a query or not).

**Proposed Work**. We present GraphFlow, a novel framework for supporting complex queries by retrieving accurate and diverse knowledge from knowledge graphs (KGs), without relying on process-level reward supervision. Inspired by GFlowNet [3], GraphFlow formulates the problem of retrieving from KGs as learning a retrieval policy that generates retrieval trajectories with probabilities proportional to their outcome rewards. Thus, the retrieval trajectory that better supports the query is retrieved with a higher probability, leading to diverse and accurate retrieval results. To achieve this, GraphFlow jointly trains the retrieval policy with a flow estimator, which assigns non-negative flow values to partial trajectories. These flow values decompose the final outcome reward across intermediate retrieval steps, thereby providing rich supervision signals without requiring explicit process-level rewards. The retrieval policy is guided by these flow values and receives process-level supervision "for free". We adopt the detailed balance objective [64] to co-train the retrieval policy and the flow estimator. To further enhance training efficiency, we introduce a local exploration strategy that reduces visits to low-reward regions of the KG. Thus, GraphFlow effectively explores high-reward regions of the retrieval space, leading to more accurate and diverse retrievals that better support complex query.

We evaluate the effectiveness of GraphFlow on the STaRK benchmark [74], which involves retrieving from text-rich KGs for real-world queries across multiple domains. Extensive experiments demonstrate that GraphFlow consistently produces high-quality and diverse retrieval results, outperforming both Process Reward Models (PRMs) and existing KG-based RAG methods. Notably, GraphFlow surpasses strong KG-based RAG baselines instantiated with GPT-4o, achieving an average improvement of 10% in both retrieval accuracy and diversity metrics. In addition, GraphFlow enjoys strong generalization capabilities and can retrieve from unseen KGs to support queries in new domains.

## 2  Preliminary and Notations

### 2.1  KG-based RAG

We denote a knowledge graph (KG) as $\mathcal{G} = \{\mathbb{V}, \mathbb{E}\}$ where $\mathbb{V}$ and $\mathbb{E}$ are the sets of nodes and edges. The node $V_i$ is associated with a short text description of an entity (e.g. $V_i$ = 'Alice'). And the edge $e_{ij}$ denotes the relationship between node $V_i$ and $V_j$. For example, $e_{ij}$ = 'daughter' represents the relationship between $V_i$ = 'Alice' and $V_j$ = 'Bob'.

**Retrieval-based Approach**. For an input query $\mathcal{Q}$, the retrieval-based method [21, 37, 13, 48, 80, 4] first encodes the texts in nodes and edges into embeddings using a pretrained LM [62]. Then, a retriever $\mathrm{Ret}(\cdot)$ is employed to retrieve a subgraph $\mathcal{G}_{\mathrm{sub}}$ from $\mathcal{G}$ [84] that can support answering $\mathcal{Q}$:

$$\max_{G_{\mathrm{sub}}} P(\mathcal{A}|\mathcal{Q}, G_{\mathrm{sub}}), \quad G_{\mathrm{sub}} = \mathrm{Ret}(\mathcal{G}). \tag{1}$$

Here $\mathcal{A}$ is the answer to the input query. Some works employ non-parametric retrievers, such as dense retriever with vector similarity [1] and Prize-Collecting Steiner Tree (PCST) algorithm [21] to retrieve from KGs. Other works train parameterized retriever models based on Multi-layer Perceptron (MLP) and Graph Neural Network (GNN) [82] to retrieve from KGs.

**Agent-based Approach**. Recent work formulates retrieval on KGs as a multi-step decision process [68, 47, 38] and employs LLM agents to search on KGs due to their superior capability in planning. For an input query $\mathcal{Q}$, the agent $\mathrm{LLM}(\cdot)$ starts from an initial node $V_0$ and searches $T$ steps incrementally in a KG to produce a retrieval trajectory $\tau = V_0 \to \cdots \to V_{T-1} \to V_T$ to support $\mathcal{Q}$:

$$\max_{\tau \in \mathcal{T}} P(\mathcal{A}|\mathcal{Q}, \tau), \quad \tau = \mathrm{LLM}(\mathcal{G}). \tag{2}$$

The inital node $V_0$ can be identified using Entity name recognition (ENR) [18] or vector similarity matching [63]. At step $t$, the searching agent expands the trajectory conditioned on the input query $\mathcal{Q}$, the partial trajectory at $t$ step $\tau_{\leq t} = V_0 \to \cdots \to V_{t-1} \to V_t$:

$$V_{t+1} \sim P_{\mathrm{LLM}}(V_{t+1}|\mathcal{Q}, \tau_{\leq t}), \quad V_{t+1} \in \mathcal{N}(V_t). \tag{3}$$

Here $P_{\mathrm{LLM}}$ is the policy instantiated by an LLM, and $\mathcal{N}(V_t)$ is the neighborhood node set of $V_t$.

### 2.2  Process Reward Models

Process Reward Models (PRMs) have emerged as a promising framework for aligning large language models (LLMs) with human preferences [85, 73, 40]. For a multi-step decision problem, denote $s \in \mathcal{S}$ as the state and $a \in \mathcal{A}$ as the action. Training a PRM requires a preference dataset $\mathcal{D} = \{(a_i^+, a_i^-, s_i) \mid i = 1, \cdots, N\}$, where $a_i^+$ and $a_i^-$ are positive and negative actions at state $s_i$, respectively. Such datasets can be constructed through human supervision [40], rule-based heuristics [55], or LLM-generated annotations [76]. The goal of PRM is to learn a scoring function $r_\theta(a, s) : \mathcal{A} \times \mathcal{S} \to \mathbb{R}$ that assigns real-valued preference scores to action–state pairs by minimizing the following objective:

$$\mathcal{L}_{\mathrm{PRM}} = -\mathrm{E}_{(a_i^+, a_i^-, s_i) \sim \mathcal{D}} \log[\sigma(r_\theta(a_i^+, s_i) - r_\theta(a_i^-, s_i))], \tag{4}$$

where $\sigma(\cdot)$ denotes the sigmoid function. Once trained, the PRM can be used to provide step-wise preference signals for LLM alignment [40, 73]. Additionally, it can directly guide the multi-step decision with a soft policy $P(s_{t+1}|s_t) \propto e^{r_\theta(s_t, a_t)}$ [7, 73, 76, 92].

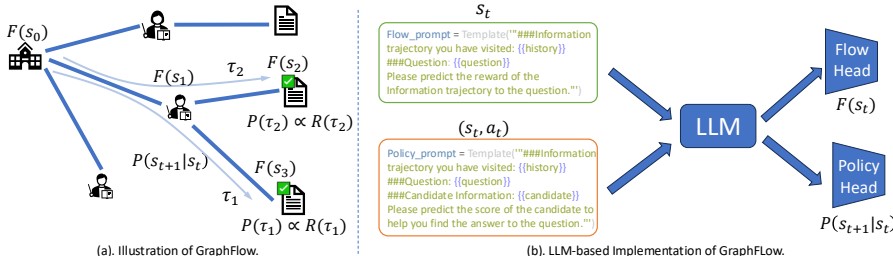

(a). Illustration of GraphFlow.  (b). LLM-based Implementation of GraphFlow.

Figure 2: An overview of the proposed GraphFlow framework. (a). GraphFlow employs a flow estimator $F(\cdot)$ to factorize the outcome reward $R(\tau)$ of a retrieval trajectory $\tau$ to flow value $F(s_t)$. The flow value guides to learn a policy $P(s_{t+1}|s_t)$ that leads to accurate and diverse retrieval results for complex queries. (b) We introduce an LLM-based implementation of GraphFlow to enhance KG-based RAG on text-rich KGs.

## 3 Method

### 3.1 KG-based RAG as Multi-step Decision

**Problem Formulation**. Given an input query $\mathcal{Q}$, our goal is to retrieve a set of $K$ target nodes $\mathbb{V}^* = \{V_i \mid i = 1, \cdots, K\}$ from a text-rich knowledge graph (KG) $\mathcal{G} = \{\mathbb{V}, \mathbb{E}, \mathbb{D}\}$ such that the associated documents $\mathbb{D}^* = \{D_i^* \mid i = 1, \cdots, K\}$ can support answering $\mathcal{Q}$. Here, $V_i \in \mathbb{V}$ is a node, $E_{ij} \in \mathbb{E}$ is an edge between $V_i$ and $V_j$ indicating their relation. Each $D_i$ denotes the textual document associated with node $V_i$ (e.g., the content of a paper).

**Agent-based Retrieval as a Multi-step Decision Process**. To effectively leverage both relational and text information in the KG, we employ the agent-based approach initiated with an LLM due to its superior text understanding and planning ability. We formulate the agent-based retrieval as a multi-step decision problem, consisting of the following components:

- **State.** The agent starts retrieval from the initial state $s_0 = (\mathcal{Q}, \{D_0\})$, where $D_0$ is the document associated with the source node $V_0$ from which the retrieval process is initiated. At step $t$, the agent arrives at node $V_t$ and the current state is defined as $s_t = (\mathcal{Q}, \{D_j\}_{j=0}^t)$, where $\{D_j\}_{j=0}^t$ is the set of documents collected along the partial retrieval trajectory $\tau_{\leq t} = V_0 \rightarrow \cdots \rightarrow V_t$.

- **Action.** Given state $s_t$, the agent selects an action $a_t \in \mathcal{A}(s_t)$, corresponding to moving from $V_t$ to a adjacent node $V_{t+1} \in N(V_t)$ along edge $E_{t,t+1}$. The agent then retrieves the documents $D_{t+1}$ associated with $V_{t+1}$.

- **Transition.** The agent transits to state $s_{t+1} = (\mathcal{Q}, \{D_j\}_{j=0}^{t+1})$. This process continues until either the document $D_{t+1}$ is deemed sufficient to support the query $\mathcal{Q}$, or a predefined maximum number of steps is reached.

- **Reward.** Upon termination, the agent receives a reward $R(\tau)$ for the retrieval trajectory $\tau$. The reward is calculated whether the document $D_T$ associated with the terminal node $V_T$ of trajectory $\tau$ can the query (i.e. $D_T \in \mathbb{D}^*$).

**Energy-based Modeling for Accurate and Diverse Retrieval**. As shown in Figure 2 (a), the goal of GraphFlow is to learn the policy $P(s_{t+1} \mid s_t)$ that can effectively retrieve accurate and diverse information from a knowledge graph (KG) to support answering an input query. To this end, we formulate the retrieval process as an energy-based distribution over trajectories:

$$P(\tau) = \prod_{t=0}^T P(s_{t+1} \mid s_t) \propto R(\tau). \tag{5}$$

The equality in Eq. 5 is due to the Markov property of the state transition.

In contrast to the objectives of prior KG-based RAG methods in Eq. 1 and Eq. 2 that maximize the likelihood of the most relevant information in KG, the objective of GraphFlow in Eq. 5 reflects the intuition that high-reward retrieval trajectories (i.e., trajectories ending in high-quality supporting documents) should be sampled more frequently. Thus, GraphFlow naturally promotes diverse yet

Table 1: Performance of retrieval accuracy of KG-based RAG methods on STaRK benchmark. Our GraphFlow outperforms baselines with higher hit rates and MRR scores. GraphFlow also surpasses strong baselines implemented with GPT-4o on most metrics.

| Method | Dataset | STaRK-AMAZON | | | STaRK-MAG | | | STaRK-PRIME | | |
|---|---|---|---|---|---|---|---|---|---|---|
| | Metric | Hit@1 ↑ | Hit@5↑ | MRR↑ | Hit@1↑ | Hit@5↑ | MRR↑ | Hit@1↑ | Hit@5↑ | MRR↑ |
| Retrieval -based | DenseRetriver | 6.10 | 15.85 | 10.61 | 24.44 | 40.23 | 32.41 | 5.43 | 13.07 | 8.99 |
| | G-Retriever | 6.10 | 11.59 | 8.54 | 24.44 | 31.95 | 28.08 | 5.43 | 8.94 | 6.95 |
| | SubgraphRAG | 8.03 | 12.43 | 9.90 | 9.30 | 25.59 | 16.11 | 4.82 | 8.00 | 6.17 |
| Agent- based (w/o Rerank) | ToG+LLaMA3 | 4.21 | 6.16 | 5.25 | 12.0 | 14.09 | 12.67 | 21.92 | 34.0 | 26.61 |
| | ToG+GPT4o | **20.67** | 41.38 | 30.90 | 23.33 | 56.67 | 36.38 | 16.67 | 39.77 | 27.02 |
| | SFT | 8.16 | 15.30 | 13.54 | 26.53 | 28.57 | 29.10 | 27.5 | 40.07 | 33.06 |
| | PRM | 20.09 | 26.25 | 28.16 | 26.05 | 28.0 | 28.52 | 21.01 | 46.72 | 31.25 |
| | **GraphFlow** | 19.63 | **44.17** | **31.66** | **29.32** | **58.64** | **41.32** | **39.84** | **71.71** | **54.58** |
| Agent- based (Rerank) | ToG+LLaMA3 | 4.21 | 6.16 | 5.25 | 12.0 | 14.09 | 12.67 | 21.92 | 34.0 | 26.61 |
| | ToG+GPT4o | 27.58 | 51.72 | 39.08 | 26.67 | 56.67 | 39.65 | **53.33** | 63.73 | 57.78 |
| | SFT | 12.24 | 30.61 | 21.54 | 27.55 | 44.89 | 36.37 | 23.75 | 52.5 | 35.98 |
| | PRM | 21.25 | 42.50 | 31.97 | 27.31 | 44.09 | 33.69 | 22.86 | 28.24 | 26.94 |
| | **GraphFlow** | **47.85** | **65.03** | **55.49** | **39.09** | **57.51** | **47.82** | 51.39 | **72.11** | **61.37** |

accurate retrieval results since the retrieval trajectories resulting in highly relevant documents are more likely to be explored. Moreover, GraphFlow also enjoys good generalization ability by avoiding strict likelihood maximization and does not overfit to a few dominant candidates.

## 3.2 Flow Estimation as Credit Assignment

A major challenge in learning the policy $P(s_{t+1} \mid s_t)$ to satisfy Eq. 5 is the lack of process-level supervision. When collecting retrieval trajectories $\tau \in \mathcal{T}$ for training, only the outcome reward $R(\tau)$ is observable, indicating whether the final retrieved document supports answering the query. Annotating the process-level reward signals for every intermediate state and action is expensive. This gives rise to the credit assignment problem [56, 91], which attributes the terminal reward of a trajectory back to the intermediate decisions. To address this, we adopt the GFlowNet framework [3], which implicitly performs credit assignment by estimating a non-negative flow value for each state.

Rather than directly maximizing the reward or value of a full trajectory, GFlowNets introduce a flow function $F(s) : \mathcal{S} \to \mathbb{R}_{\geq 0}$ for each intermediate state $s$. The learning objective enforces a local consistency constraint between transitions, which is known as the detailed balance condition:

$$F(s_t) \cdot P(s_{t+1} \mid s_t) = F(s_{t+1}) \cdot P_{\mathrm{B}}(s_t \mid s_{t+1}), \tag{6}$$

where $P(s_{t+1} \mid s_t)$ is the forward policy we want to obtain, and $P_{\mathrm{B}}(s_t \mid s_{t+1})$ is the backward policy. When this condition holds for all transitions, the retrieval trajectory induced by the policy $P(s_{t+1} \mid s_t)$ satisfies the objective in Eq. 5, leading to diverse and accurate retrieval results on KGs.

While alternative GFlowNet objectives, such as trajectory balance [52] or subtrajectory balance [51], can also promote diversity, they require computation over entire trajectories or sub-trajectories. In KG-based RAG, the retrieval trajectory involves multi-hop transitions, and each node is associated with long documents. These objectives are computation-intensive and often lead to out-of-memory (OOM) issues. To ensure scalability and efficiency, we thus adopt the detailed balance objective that operates on state transitions.

**Detailed Balance with Local Exploration**. Enforcing the detailed balance condition globally across all transitions in a knowledge graph (KG) is computationally inefficient, since the vast state space makes many nodes and transitions unreachable during training. To address this, we introduce a local exploration strategy that focuses the detailed balance objective on the neighborhoods of states observed in sampled trajectories.

For a retrieval trajectory $\tau = V_0 \to \cdots \to V_T$ with reward $R(\tau)$, we apply local exploration to each non-terminal state $s_t = (\mathcal{Q}, \{D_j\}_{j=0}^t)$ where $t \neq T$. Specifically, the agent takes an exploratory action $a_t' \in \mathcal{A}(s_t)$ that moves from node $V_t$ to a neighboring node $V_{t+1}' \in \mathcal{N}(V_t)$ different from the original next node $V_{t+1}$. This results in a new exploratory state $s_{t+1}' = (\mathcal{Q}, \{D_j\}_{j=0}^t \cup \{D_{t+1}'\})$, corresponding to the partial trajectory $\tau_{\leq t+1}' = V_0 \to \cdots \to V_t \to V_{t+1}'$.

Table 2: Performance of retrieval diversity of KG-based RAG methods. GraphFlow retrieves more correct documents to support queries with high diversity.

| Method | Dataset | STaRK-AMAZON | | STaRK-MAG | | STaRK-PRIME | |
|---|---|---|---|---|---|---|---|
| | Metric | R@20↑ | D-R@20↑ | R@20↑ | D-R@20↑ | R@20↑ | D-R@20↑ |
| Retrieval -based | DenseRetriver | 13.63 | 13.63 | 41.80 | 41.80 | 13.92 | 13.92 |
| | G-Retriever | 5.35 | 5.35 | 25.37 | 25.37 | 6.75 | 6.75 |
| | SubgraphRAG | 6.53 | 6.53 | 27.83 | 26.95 | 6.49 | 6.49 |
| Agent -based | ToG+LLaMA3 | 2.61 | 2.61 | 6.77 | 6.77 | 33.84 | 33.84 |
| | ToG+GPT4o | 25.81 | 23.70 | 48.03 | 47.71 | 54.35 | 54.35 |
| | SFT | 25.22 | 24.97 | 37.48 | 35.90 | 47.72 | 45.36 |
| | PRM | 35.72 | 18.94 | 36.73 | 36.73 | 45.97 | 45.97 |
| | **GraphFlow** | **36.15** | **36.15** | **57.18** | **57.18** | **79.71** | **79.59** |

By performing $k$ such explorations, we generate $k$ exploratory actions $\{a'_{t,1}, \cdots, a'_{t,k}\}$ and their resulting states $\{s'_{t+1,1}, \cdots, s'_{t+1,k}\}$. With the ground-truth next state denoted as $s'_{t+1,0} = s_{t+1}$, we obtain $k+1$ transitions from $s_t$ to candidate next states for optimizing the detailed balance objective. The forward policy is then defined as $P(s_{t+1} = s'_{t+1,i}|s_t) = \frac{e^{r_\theta(s_t, a'_{t,i})}}{\sum_{i=0}^{k} e^{r_\theta(s_t, a'_{t,i})}}$. Here $r_\theta(s, a)$ is a learned process reward function parameterized by a neural network with parameters $\theta$. Since the retrieval process is inherently irreversible (i.e., backtracking is disallowed), we follow prior work [22] and set the backward policy $P_B(s_t \mid s_{t+1}) = 1$ in Eq. 5. We yield the following objective for state $s_t$ by taking the $\log$ function to both sides of Eq. 5:

$$
\begin{aligned}
\mathcal{L}_{\text{DBLE}}(s_t) &= \sum_{i=0}^{k} [\log F(s_t) - \log F(s'_{t+1,i}) + \log P(s_{t+1} = s'_{t+1,i}|s_t)]^2 \\
&= \sum_{i=0}^{k} [\log F(s_t) - \log F(s'_{t+1,i}) + r_\theta(s_t, a'_{t,i}) - \log \sum_{i=0}^{k} e^{r_\theta(s_t, a'_{t,i})}]^2.
\end{aligned}
\tag{7}
$$

**Boundary Condition**. We impose boundary conditions on the initial and terminal states to ensure proper propagation of flow values along the retrieval trajectory: $\log F(s_0) = \log F(s_T) = 1$. Here $s_0$ is the initial state and $s_T$ is the terminal state. The reason is that we only collect retrieval trajectories that reach target documents during model training. With such boundary conditions, we ensure that the total incoming and outgoing flow is consistent across the trajectory and enable the flow estimator to correctly distribute the outcome reward of the terminal state to the intermediate states.

**Termination Condition**. To allow the retrieval policy $P(s_{t+1} \mid s_t)$ to decide when to stop, we introduce a special self-loop action that retrieves the current node again. At each step, this action is included among the candidate actions in Eq. 7. Hence, Eq. 7 can also be applied for the terminal state. If the policy chooses to retrieve the current document (i.e., selects the self-loop), the trajectory is terminated, indicating that the current document is relevant to the query. Otherwise, the policy continues to explore the KG. This mechanism enables the agent to adaptively determine when to stop retrieval based on its experience, rather than relying on a fixed number of steps.

**Difference Between GraphFlow, SFT, and PRM**. SFT and PRM learn the retrieval policy by treating the action leading to the ground-truth next state $s_{t+1}$ as a positive sample, and exploratory actions leads to $\{s'_{t+1,1}, \cdots, s'_{t+1,k}\}$ as negative ones, akin to behavior cloning [70]. GraphFlow generalizes this by learning state-dependent flow values $F(s)$ and factorizes the outcome reward via Eq. 7. When setting $\log F(s_t) = 1$ and $\log F(s'_{t+1,i}) = 0$, GraphFlow reduces to behavior cloning. However, such as a hard objective limits generalization and cannot learn a policy leading to diverse and accurate retrieval results for complex queries.

### 3.3 Instantiating GraphFlow with LLMs

We implement GraphFlow with an LLM due to its ability of text understanding and decision-making, as shown in Figure 2 (b). The state and state-action pair are decorated with a flow prompt and policy prompt template, which are encoded using a shared LLM. The embeddings of the final tokens are used as representations of the state and the state–action pair, respectively. On top of the shared

Table 3: Quantitative retrieval quality of different KG-based retrieval methods.

| Method | STaRK-Amazon | | STaRK-MAG | | STaRK-PRIME | |
|---|---|---|---|---|---|---|
| | Step-$\Delta_{Seper}$ ↑ | Answer-$\Delta_{Seper}$ ↑ | Step-$\Delta_{Seper}$ ↑ | Answer-$\Delta_{Seper}$ ↑ | Step-$\Delta_{Seper}$ ↑ | Answer-$\Delta_{Seper}$ ↑ |
| ToG+GPT-4o | $0.031 \pm 0.109$ | $0.092 \pm 0.128$ | $0.041 \pm 0.172$ | $0.065 \pm 0.150$ | $0.065 \pm 0.125$ | $0.148 \pm 0.165$ |
| ToG+LLaMA3 | $0.010 \pm 0.118$ | $0.046 \pm 0.149$ | $0.068 \pm 0.108$ | $0.105 \pm 0.146$ | $0.009 \pm 0.102$ | $0.021 \pm 0.106$ |
| SFT | $0.079 \pm 0.151$ | $0.141 \pm 0.160$ | $0.035 \pm 0.101$ | $0.084 \pm 0.095$ | $0.062 \pm 0.132$ | $0.158 \pm 0.183$ |
| PRM | $0.029 \pm 0.089$ | $0.071 \pm 0.112$ | $0.037 \pm 0.117$ | $0.060 \pm 0.115$ | $0.057 \pm 0.106$ | $0.131 \pm 0.174$ |
| G-Retriever | — | $0.024 \pm 0.110$ | — | $0.012 \pm 0.089$ | — | $0.029 \pm 0.117$ |
| SubgraphRAG | — | $0.021 \pm 0.093$ | — | $0.039 \pm 0.076$ | — | $0.046 \pm 0.083$ |
| GraphFlow | $\mathbf{0.097 \pm 0.158}$ | $\mathbf{0.219 \pm 0.257}$ | $\mathbf{0.081 \pm 0.137}$ | $\mathbf{0.145 \pm 0.112}$ | $\mathbf{0.091 \pm 0.147}$ | $\mathbf{0.206 \pm 0.192}$ |

encoder, we employ two separate multi-layer perceptrons (MLPs) as the policy head and the flow head, respectively. The policy head predicts the forward transition probability, while the flow head estimates logarithm of the flow value of the state. During model training, we apply LoRA [23] to inject learnable adapters into the frozen backbone of the LLM, and update the parameters of the flow head and the policy head. This design enables joint optimization of policy learning and flow estimation in a parameter-efficient manner, while also capturing rich contextual information through the LLM encoder. We present detailed implementation in Supplementary Material due to space limit.

# 4 Experiment

## 4.1 Dataset

We employ the STaRK [74] benchmark to validate the retrieval quality of the proposed GraphFlow to support complex queries. STaRK is a recently proposed benchmark designed to evaluate the retrieval performance of KG-based RAG methods on text-rich KGs spanning three domains:

- **STaRK-AMAZON** is an e-commerce KG where the nodes contain detailed product information and the edges denotes the properties of products and co-purchase between products. The retrieval task is to retrieve the diverse products to satisfy the recommendation query.

- **STaRK-MAG** is an academic graph constructed based on OGB [24] and Microsoft Academic Graph [66]. The nodes contain author information, institute, and publications. The retrieval task is to address academic queries such as paper searching.

- **STaRK-PRIME** is a biomedical KG where the nodes are associated with the detailed description of drugs, disease, genes, and pathways, and the edges are their relationship. The retrieval task is to address the biomedical query.

The StaRK benchmark challenges KG-based RAG methods by complex queries corresponding to diverse retrieval targets and fusion of text and structure information that complicates accurate retrieval.

## 4.2 Baseline and Evaluation Metrics

**Baseline**. We choose representative retrieval-based and agent-based baselines with explicit retrieval results (i.e., the retrieved node index) on the STaRK benchmark [74]. All detailed implementations are shown in Supplementary Material due to space limit.

For retrieval-based baselines, we consider **Dense-Retriever** [30], **G-Retriever** [21], and **SubgraphRAG** [37]. Dense-Retriever is implemented with SentenceBERT [62] to encode both questions and the documents of KG nodes into dense embeddings and retrieve the documents with top vector similarity. G-Retriever employs the Prize-Collecting Steiner Tree (PCST) [2] algorithm to extract a subgraph from KGs relevant to the query. Since computing PCST on STaRK benchmark is infeasible, we follow the hybrid setting [33, 34] that first identifies a source node in KG via Dense-Retriever and only computes PCST around the ego-graph up to 2 hops around the identified node. We also adopt the same hybrid setting for other baselines to ensure computational feasibility on STaRK. SubgraphRAG integrates a learnable subgraph retrieval module to retrieve from KG.

For agent-based methods, we consider **ToG** [68], **SFT**, and **PRM** [40] as baselines. ToG employs an LLM agent to search from the KG to retrieve supporting documents. We instantiate ToG using both LLaMA3-8B-Instruct and GPT-4o as backbone models, denoted as ToG+LLaMA3 and ToG+GPT4o,

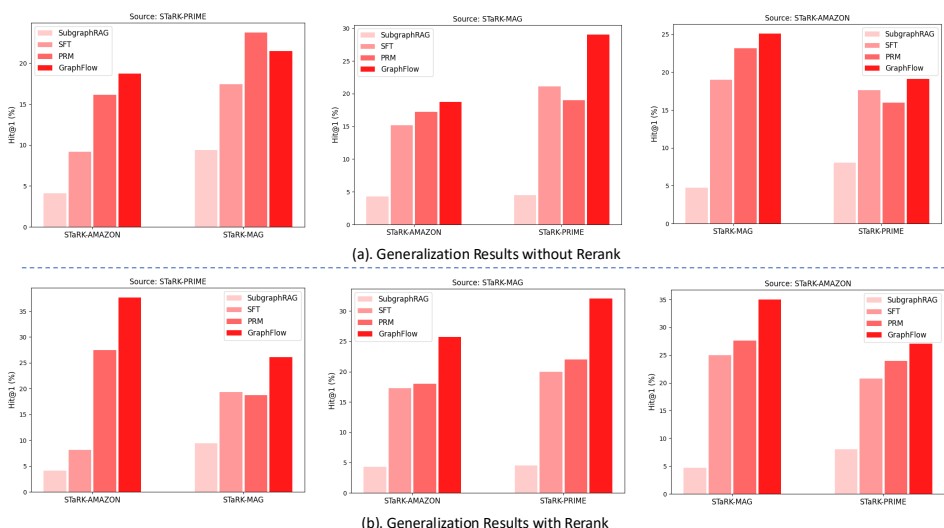

(a). Generalization Results without Rerank

(b). Generalization Results with Rerank

Figure 3: Generalization Performance of KG-based RAG methods. GraphFlow shows superior cross-domain generalization performance, especially under the rerank setting (best viewed in color).

respectively. SFT and PRM are two popular approaches that fine-tune the LLM agent to enhance RAG in recent works [76, 12, 25]. SFT, PRM, and GraphFlow are deployed with LLaMA3-8B-Instruct [17]. For agent-based methods, we use the agent to rerank all the retrieved results.

**Evaluation Metrics**. All KG-based RAG methods retrieve every input query 20 times and generate 20 retrieval results for diversity and accuracy evaluation following the standard setting in STaRK [74]. We employ the following metrics to evaluate the retrieval performance. Hit@k denotes whether the ground truth is retrieved in the top-k results. We employ **Hit@1 and Hit@5** to measure the retrieval precision of the different KG-based RAG methods. **Mean Reciprocal Rank (MRR)** measures the average of reciprocal ranks of the first ground-truth item in the retrieval results and encourages the ground-truth item to be retrieved in a higher rank. **Recall@k (R@k)** is a standard metric to measure the percentage of ground-truth items that appear in the top-$k$ retrieved results. We employ Recall@20 (R@20) for evaluation. **De-duplicate Recall@k (D-R@k)** measures the percentage of *unique* ground-truth items that appear in the top-$k$ retrieved results. This metric is used to evaluate the diversity of the correctly retrieved results. We use De-duplicate Recall@20 (D-R@20).

## 4.3   Main Results

**Accuracy**. Table 1 presents the retrieval accuracy of various KG-based RAG methods on STaRK.

GraphFlow consistently outperforms other KG-based RAG approaches on most metrics. In particular, it achieves higher Hit rates and MRR scores than the strong baseline ToG+GPT-4o with an average 10% improvement in retrieval accuracy. Interestingly, ToG's performance is highly sensitive to the choice of backend model. When instantiated with LLaMA3-8B, ToG shows a significant drop in performance compared to using GPT-4o. Additionally, rerank has no effect in the ToG+LLaMA3-8B setup, as all retrieved results receive equally high scores, leaving the ranking unchanged.

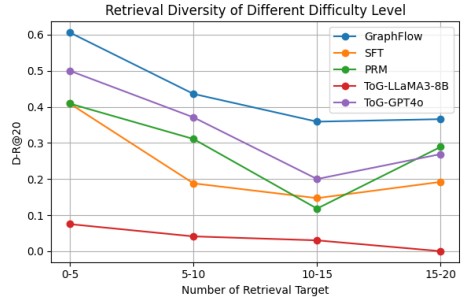

Figure 4: GraphFlow shows improved retrieval diversity on different difficulty levels of retrieval queries on STaRK-PRIME.

Two finetuned agent-based baselines, SFT and PRM, outperform ToG without finetuning, but still fall short of GraphFlow. While PRM training can leverage curated preference datasets with fine-grained process-level rewards, such labeling is prohibitively expensive. Instead, GraphFlow achieves high-quality retrieval with only outcome rewards of retrieval trajectories. For retriever-based approaches, DenseRetriever, G-Retriever, and SubgraphRAG show moderate performance but are generally inferior to agent-based

methods. Overall, retriever-based methods remain more lightweight but trail behind agent-based approaches in retrieval accuracy.

**Diversity**. Table 2 reports the retrieval diversity of different KG-based RAG methods on the STaRK benchmark. We evaluate both Recall@20 (R@20) and its de-duplicated variant (D-R@20), which better captures retrieval diversity. GraphFlow achieves the highest retrieval diversity across all datasets, outperforming both retriever-based and agent-based baselines. Its results not only match more ground-truth contents but also avoid redundancy. Notably, GraphFlow exceeds the strongest baseline (ToG+GPT-4o) by a large margin on the STaRK-PRIME dataset, highlighting its ability to retrieve results that are both relevant and diverse. Compared with PRM and SFT, GraphFlow also demonstrates superior diversity. In contrast, retriever-based methods retrieve less diverse content and cover fewer retrieval targets.

## 4.4 Quantifying Retrieval Quality

We further employ the Seper score ($\Delta_{Seper}$) [9] to quantify the retrieval quality of different KG-based retrieval methods. The Seper score is a recently proposed metric for evaluating retrieval utility by measuring semantic perplexity reduction after retrieval: $\Delta_{Seper} = p_M(a|q, d) - p_M(a|q)$. Here, $q$ is the question, $d$ is the document associated with the retrieval item, and $M$ is an LLM used for question answering. In our case, we use LLaMA3–8B–Instruct to instantiate $M$ to keep consistent with the retrieval model. Since there is no ground-truth answer for the questions in the STaRK benchmark, we use the title or summarized description of the ground-truth retrieval item as $a$. We design the following metrics for comprehensive evaluation and report their mean and standard deviation (std).

- Step-$\Delta_{Seper}$: the Seper score that quantity the retrieval quality of intermediate retrieval.
- Answer-$\Delta_{Seper}$: the Seper score that quantity the retrieval quality of the final result.

As shown in Table 3, GraphFlow consistently achieves higher Step-$\Delta_{Seper}$ and Answer-$\Delta_{Seper}$ than all the baselines, demonstrating stronger information utility during retrieval. These results further confirm that GraphFlow can significantly improve the information utility when retrieving from text-rich KGs. Notice that all the methods have high variance in Step-$\Delta_{Seper}$ and Answer-$\Delta_{Seper}$. The reason is the high variance in natural language entailment when calculating Seper scores. Moreover, we observe that some retrieval samples have negative Seper scores for all methods, indicating a negative impact on question answering when retrieving bad contents.

## 4.5 Further Discussion

**Cross-domain Generalization**. Figure 3 reports the cross-domain generalization ability of different KG-based RAG methods. We use Hit@1 to evaluate the retrieval accuracy Compared with SubgraphRAG using a small model for retrieval, SFT, PRM, and GraphFlow that finetune the LLM show better cross-domain generalization ability due to the *over-parameterization* [35, 15]. GraphFlow demonstrates superior cross-domain generalization, since it avoids from likelihood maximization objectives used by SFT and PRM. Instead, GraphFlow adaptively assigns the outcome reward of the retrieval trajectory to the flow values of intermediate states and guides the retrieval policy, leading to better generalization ability. More results are shown in Supplementary Material due to space limit.

**Performance on Hard Cases**. We categorize the retrieval queries with different numbers of retrieval targets into 4 difficulty levels. Figure 4 shows the D-R@20 scores of KG-based RAG methods on retrieval queries on STaRK-PRIME at different levels. GraphFlow outperforms the other agent-based approaches by a large margin by covering more diverse and accurate retrieval targets, especially on the hard cases containing more than 15 retrieval targets. The performance on hard cases shows the superior performance of GraphFlow in retrieving more relevant and diverse results. More results of hard cases are shown in Supplementary Materials due to space limit.

## 5  Related Work

**KG-based RAG**. Knowledge graphs (KGs) are widely used as knowledge sources in retrieval-augmented generation (RAG) [20**?** ] systems to enhance large language models (LLMs) with both relational and textual information for answering complex queries [14, 13, 8, 75]. A core challenge in

KG-based RAG lies in retrieving relevant knowledge from KGs in response to a given query. Recent methods addressing this problem can be broadly categorized into two approaches. Retrieval-based methods [80, 53, 21, 61, 93] leverage pretrained language models to encode the textual content in KGs into embeddings and use small models, such as MLP and GNN, to retrieve relevant information. In contrast, agent-based methods [68, 88, 50, 47, 45] employ LLMs as agents that iteratively traverse the KG to locate supporting evidence. While both paradigms have shown promise in knowledge graph question answering (KGQA) [37], their effectiveness in retrieving diverse and high-quality candidates for complex queries remains limited. Furthermore, complex queries often require retrieval from text-rich KGs, necessitating the joint consideration of both relational structure and text content. Although recent studies [50, 34] have begun to explore this setting and tackle complex queries, enhancing the diversity and accuracy of KG-based RAG is still underexplored.

**Process Reward Models**. Process Reward Models (PRMs) [40, 32] have shown great promise in guiding LLMs with process supervision and have been adopted in many domains such as complex reasoning [7], alignment [58], and planning [6]. The key to PRMs is to construct a preference dataset with process supervision [60]. Previous works obtain the process supervision from human feedback and LLM evaluation. However, fine-grained process-level supervision is expensive for KG-based RAG due to the potentially vast search space of KGs and the difficulty of accessing the intermediate state during retrieval. Although early explorations are made to use PRM to guide the retrieval process of RAG on unstructured knowledge bases [39, 65, 25, 76], they still need a preference dataset with process-level supervision. How to guide the retrieval process of KG-based RAG on structured KGs without process supervision data is still challenging.

**GFlowNet**. GFlowNet [3] aims to sample diverse and high-quality candidates from an unnormalized density and has received increasing attention in sampling from discrete and vast spaces [16, 46, 11, 10, 49]. The goal of GFlowNet is to learn a policy that can lead to the terminal states with the likelihood in proportion to their rewards [86]. Some objects are proposed to optimize GFlowNet by regularizing the state flows and their transitions [69, 52, 51]. Recently, GFlowNet has also been introduced to improve the generative performance of LLMs and diffusion models by promoting diversity in the decoding process [71, 22, 81, 29, 43] . Differently, our work focuses on aligning the retrieval results of KG-based RAG with the knowledge required for real-world queries by estimating the state flow in multi-step retrieval. Moreover, we introduce a local exploration strategy to avoid visiting less-valued states, thus efficiently optimizing the detailed balance.

## 6 Conclusion

We introduce GraphFlow, a novel framework that enhances existing KG-based RAG methods by enabling accurate and diverse retrieval from text-rich KGs. By jointly optimizing a retrieval policy and a flow estimator via a detailed balance objective, GraphFlow effectively aligns the retrieval process with query-specific knowledge demands without explicit process-level reward. Extensive evaluation on the STaRK benchmark demonstrates that GraphFlow not only surpasses strong baselines deployed with GPT-4o, but also generalizes well to unseen KGs. These findings underscore the effectiveness and robustness of GraphFlow in supporting complex queries using textual and structured knowledge. Our future work will incorporate causality into KG-based RAG to improve the reasoning ability of LLMs [44, 5, 83], reduce forgetting [41, 42], and explore their scientific applications [78].

**Acknowledgments and Disclosure of Funding**

This work is supported by the UKRI grant: Turing AI Fellowship EP/W002981/1. This work is jointly supported by the Shanghai Municipal Science and Technology Major Project and Shanghai Artificial Intelligence Laboratory. The authors would like to thank PCs, ACs, and all the reviewers for their insightful comments and suggestions.

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

## A   Introduction of GFlowNet

Generative Flow Networks (GFlowNets) [3] aim to learn a stochastic policy that generates objects $x \in \mathcal{X}$ through sequential decisions, such that the marginal probability of generating $x$ is proportional

to a reward function $R(x) > 0$. Given a complete trajectory $\tau = (s_0, a_1, s_1, \ldots, a_T, s_T = x)$ that terminates in object $x$, the forward trajectory probability is:

$$P_F(\tau) = \prod_{t=0}^{T-1} P_F(a_{t+1}|s_t)$$

and the backward probability (used to reverse the trajectory) is:

$$P_B(\tau) = \prod_{t=1}^{T} P_B(a_t|s_t)$$

**Trajectory Balance (TB) Loss [52].**. The Trajectory Balance objective ensures the ratio of forward to backward probability matches the reward:

$$\frac{P_F(\tau)}{P_B(\tau)} = \frac{R(x)}{Z} \quad \Longleftrightarrow \quad \log P_F(\tau) - \log P_B(\tau) = \log R(x) - \log Z$$

where $Z$ is the global partition function. The loss function is then defined as:

$$\mathcal{L}_{\text{TB}} = (\log P_F(\tau) - \log P_B(\tau) - \log R(x) + \log Z)^2$$

In practice, $\log Z$ is treated as a learnable scalar parameter.

**Subtrajectory Balance (SubTB) Loss [51].**. To enable learning from partial trajectories, the Subtrajectory Balance loss generalizes TB to arbitrary subpaths. For any subtrajectory $\tau_{i:j} = (s_i, a_{i+1}, \ldots, s_j)$ from state $s_i$ to $s_j$, the balance condition becomes:

$$\frac{P_F(\tau_{i:j})}{P_B(\tau_{i:j})} = \frac{Z(s_j)}{Z(s_i)} \quad \Longleftrightarrow \quad \log P_F(\tau_{i:j}) - \log P_B(\tau_{i:j}) = \log Z(s_j) - \log Z(s_i)$$

This leads to the Subtrajectory Balance loss:

$$\mathcal{L}_{\text{SubTB}} = (\log P_F(\tau_{i:j}) - \log P_B(\tau_{i:j}) - \log Z(s_j) + \log Z(s_i))^2$$

Here, $Z(s)$ denotes the flow or partition function at state $s$, typically parameterized by a neural network as $F_\phi(s) = \log Z(s)$. SubTB enables more flexible and sample-efficient training, especially for long-horizon generation tasks. However, directly implementing GFlowNet on KG-based RAG faces several challenges. First, the objectives such as Trajectory balance and sub-trajectory balance are computed on the whole trajectories, leading to computational burden in KG-based RAG where entities are associated with long texts. Second, many states and transitions in KGs are less-valued and not visited, making the traditional GFlowNet objective inefficient. Second, the discrete and symbolic nature of KGs poses difficulty in defining state transitions and flow dynamics, especially when integrating pretrained language models to interpret semantic relevance. These factors collectively make it challenging to directly apply GFlowNet to KG-based retrieval without significant adaptations in trajectory design, reward shaping, and exploration strategy.

# B    Implementation of GraphFlow with LLMs

**Model Architecture**. We use LLaMA3-8B-Instruct as the backbone LLM to implement GraphFlow. Specifically, we first employ the following flow prompt template to wrap the retrieval trajectory $\tau_{\leq t}$ at state $s_t$ into a text sequence for flow estimation.

> ###Information trajectory you have visited: {history}
> ###Question: {question}
> Please predict the reward of the Information trajectory to the question:

Here {history} is the concatenation of documents of previously visited entities. {question} is the input complex query. The backbone LLM encodes the above wrapped text sequence. The embedding of the last token is treated as the representation of the wrapped sequence used for flow estimation. We employ a 1-layer MLP as the flow head, which receives the representation of the wrapped sequence and outputs the log value of the estimated flow $\log s_t$.

Then we employ the following policy prompt template to warp the retrieval trajectory $\tau_{\leq t}$ at state $s_t$ into a text sequence for policy learning.

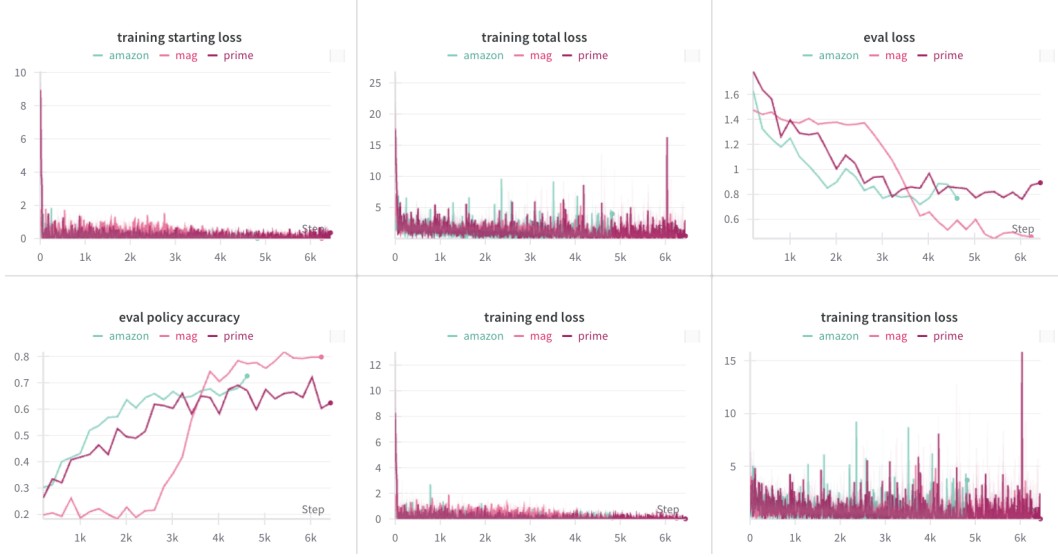

Figure 5: Training dynamics of GraphFlow.

###**Information trajectory you have visited:** {history}
###**Question:** {question}
###**Candidate Information:** {candidate}
Please predict the score of the candidate to help you find the answer to the question:

Here {history} is the concatenation of documents of previously visited entities. {question} is the input complex query. And {candidate} is one action $a_t$ that leads to the next state $s_{t+1}$. The backbone LLM encodes the above wrapped text sequence. The embedding of the last token is treated as the representation of the wrapped sequence used for learning $P(s_{t+1}|s_t)$. Specifically, we employ a 1-layer MLP with a ReLU function to parameterize $\sigma_\theta(s_t, a_t)$. The forward policy $P(s_{t+1}|s_t)$ is calculated as below:

$$P(s_{t+1}|s_t) = \frac{\sigma_\theta(s_t, a_t)}{\sum_{a_t} \sigma_\theta(s_t, a_t)}. \tag{8}$$

**Training Configuration**. we apply LoRA [23] to inject learnable adapters into the frozen backbone of the LLM, and update the parameters of the flow head and the policy head. This design enables joint optimization of policy learning and flow estimation in a parameter-efficient manner, while also capturing rich contextual information through the LLM encoder. The parameters of these modules are trained by optimizing the detailed balance with local exploration (DBLE) objective:

$$
\begin{aligned}
\mathcal{L}_{\text{DBLE}}(s_t) &= \sum_{i=0}^{k} [\log F(s_t) - \log F(s'_{t+1,i}) + \log P(s_{t+1} = s'_{t+1,i}|s_t)]^2 \\
&= \sum_{i=0}^{k} [\log F(s_t) - \log F(s'_{t+1,i}) + r_\theta(s_t, a'_{t,i}) - \log \sum_{i=0}^{k} e^{r_\theta(s_t, a'_{t,i})}]^2.
\end{aligned}
\tag{9}
$$

**Experimental Settings**. To facilitate training the LoRA module and the flow head and the policy head on the STaRK benchmark, we first collect training dataset consisting of transitions between states. For a given question $\mathcal{Q}$ with the set of ground truth retrieval entities $V_T$ in the training set, we first identify the initial entity $V_0$ using vector similarity between the embedding of $\mathcal{Q}$ and $V_0$. Then, we sample the trajectory $\tau_{\leq T} = V_0 \to \cdots \to V_T$ staring from $V_0$ and ending at $V_T$. We collect the all the transitions between $s_t$ to $s_{t+1}$ in the example the trajectory $\tau_{\leq T} = V_0 \to \cdots \to V_T$

Table 4: Parameters of GraphFlow training on STaRK benchmark.

| | STaRK-AMAZON | STaRK-MAG | STaRK-PRIME |
|---|---|---|---|
| Accumulation steps | | 2 | |
| alpha | | 16 | |
| batch_size | | 1 | |
| num_gpu | | 8 | |
| depth_cutoff | | 6 | |
| doc_cutoff | | 400 | |
| eval_ratio | | 0.8 | |
| eval_step | | 100 | |
| lora_dropout | | 0.05 | |
| lr | | 1.00E-05 | |
| max_length | | 1024 | |
| n_epochs | | 1 | |
| num_exploration | | 4 | |
| r | | 32 | |
| window_size | | 3 | |

Table 5: We provide data statistics of STaRK. The statistics are from the STaRK benchmark [74].

| | entity type | relation type | avg. degree | entities | relations | tokens |
|---|---|---|---|---|---|---|
| STARK-AMAZON | 4 | 5 | 18.2 | 1,035,542 | 9,443,802 | 592,067,882 |
| STARK-MAG | 4 | 4 | 43.5 | 1,872,968 | 39,802,116 | 212,602,571 |
| STARK-PRIME | 10 | 18 | 125.2 | 129,375 | 8,100,498 | 31,844,769 |

to implement local exploration as introduced in Section 3.2 in the main paper. For every training step, we construct mini-batch of traditions between states to calculate the loss in Eq. 9. The training dynamic is shown in Figure 5. Here, training transition loss is calculated using the transition between non-terminal states. And training starting loss and training end loss are calculated using boundary condition $F(s_0) = F(s_T) = 0$. Training total loss and eval loss are calculated on all the transitions between states on the training and evaluation dataset. Eval policy accuracy is the accuracy of policy $P(s_{t+1}|s_t)$ on the evaluation dataset. We training GraphFlow on these dataset for one epoch, other important parameters are shown in Table 4.

## C    Implementations of Baselines

To the best of our knowledge, few KG-based RAG methods are implemented on the text-rich STaRK benchmark. Instead, many KG-based RAG methods employ simple KGQA datasets such as CWQ, WEBQSP. Thus, we choose representative retrieval-based and agent-based baselines with explicit retrieval results (i.e., the retrieved node index) on the STaRK benchmark [74]. We provide the implementation details of the used baseline methods as below.

Dense-Retriever is implemented with SentenceBERT [62] to encode both questions and the documents of KG nodes into dense embeddings and retrieve the documents with top vector similarity. We choose SentenceBERT as the text document to be consistent with prior works [21], where SentenceBERT is used to encode the text information in KGs. Although STaRK benchmark provide the pre-processed text embedding of entities and relationships in KGs using text-embedding-ada-002 model, we find the inconsistency between the entities IDs and the entities embeddings. Some entities in KGs are not converted into embeddings. Thus, we rerun the encoding model using SentenceBERT to obtain the full entities embeddings. After encoding the text information into embeddings, we employ the vector similarity between the question embedding and text embeddings for retrieval. We evaluate the retrieval performance on top 20 retrieval results.

G-Retriever [21] is a two-stage method for KG-based RAG. It first employs the Prize-Collecting Steiner Tree (PCST) [2] algorithm to retrieve a subgraph from KGs relevant to the query. Then, the

retrieved subgraph is encoded into the token space of LLM using a GNN for question answering (QA). To further improve the QA performance, G-Retriever also applies LoRA module to fine-tune LLM. Since we focus on the evaluating the retrieval performance of different KG-based RAG methods, we do not fine-tune the GNN and LLM for QA. To make PCST algorithm feasible on STaRK benchmark, we adopt a hybrid approach that first identify the 20 seed nodes and implement the PCST algorithm to extract the subgraphs around 2-hop ego graph around the seed nodes. We drop the seed nodes with dense neighborhoods to avoid computation overhead [33, 34].

SubgraphRAG [37] integrates a learnable subgraph retrieval module to retrieve from KGs. Since training the subgraph retrieval module on the STaRK benchmark is infeasible, we employ the ego-graph setting similar to G-Retriever. We identify the up-to 2 hop neighbor hood graph around the seed node to construct the training and testing set for SubgraphRAG. We also drop the the seed node that has dense neighborhood to avoid computation overhead. This ego-graph setting is also employed to construct the test set for the other KG-based RAG models. We follow the default setting of SubgraphRAG to reproduce it on STaRK benchmark.

ToG [68] employs an LLM agent to search from the KG to support KG-based question anwsering. ToG is implemented with frozen LLMs by prompt engineering instead of fine-tuning. Specifically, ToG employs tree-based search [79] to transverse the KG and search the relevant information for KG-based QA. Since we focus on evaluating the retrieval performance of KG-based RAG models, we modify ToG to retrieve the relevant document at each searching steps instead of incorporating the retrieved document to update the question answering results. Since running ToG on the whole KGs in STaRK is infeasible, we identify the seed node for ToG searching using vector similarity and constrain the searching area around the 2-hop neighborhood of the seed node. We instantiate ToG using both LLaMA3-8B and GPT-4o as backbone models, denoted as ToG+LLaMA3 and ToG+GPT4o, respectively.

We also implement SFT and PRM as two fine-tuning baselines build upon ToG and LLaMA3-8B-Instruct. We use the sample training dataset to train ToG using SFT and PRM as GraphFlow for a fair comparison. We employ the **TRL** (Transformer Reinforcement Learning) package to fir SFT and PRM fine-tuning. We apply LoRA funetuning to improve the efficiency.

**Other potential baselines but hard to implement on STaRK**. There are alternative KG-based RAG baseline methods for evaluation. However, we find it hard to implemented these baseline on STaRK, mostly due to the compatibility issues. We list some examples as below.

QAGNN [80] is designed for improving the QA performance on KG-based QA task. Although its retrieval performance is reported on STaRK benchmark, detailed implementation code on STaRK is not publicly available. Although recent concurrent work [34, 33] tried to implement QAGNN on STaRK, the reported performances of QAGNN diverge from the reported results on STaRK benchmark.

RoG [47] adopts similar approach as it finetunes the LLM to search from KGs. It first employs an LLM to generate retrieval trajectories for the input queries and use the generated retrieval trajectories to construct a training dataset to fine-tune the retrieval agent by SFT. However, we find that LLM usually generate invalid retrieval trajectory, leading to low quality training datasets for SFT fine-tuning. Thus, we finetune the retrieval agent using the valid retrieval trajectories by SFT in the main paper.

ToG-2.0 [50] is a recently proposed method to retrieve from the structured database and unstructured database. The key to ToG-2.0 is to identify the topic entitiies for a given questions. However, the implementation of topic entity recognition is absent, making it difficult to reproduce ToG-2.0 on STaRK benchmark.

HybridRAG [33], Mixture of RAG [34], and KAR [75] are recent pre-prints on Arxiv focusing on retrieving from text-rich KGs. However, their codes are not available yet, making it difficult for us to reproduce these methods.

# D    More results of Cross-domain Generalization

We show more generalization performance in terms of Hit@5 in Figure 6.

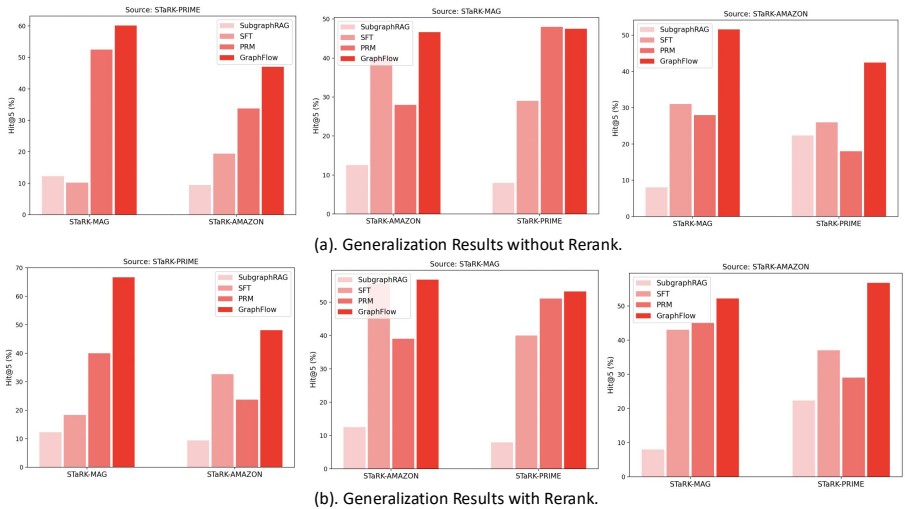

Figure 6: Generalization Performance (Hit@5) of KG-based RAG methods. GraphFlow shows superior cross-domain generalization performance, especially under the rerank setting (best viewed in color).

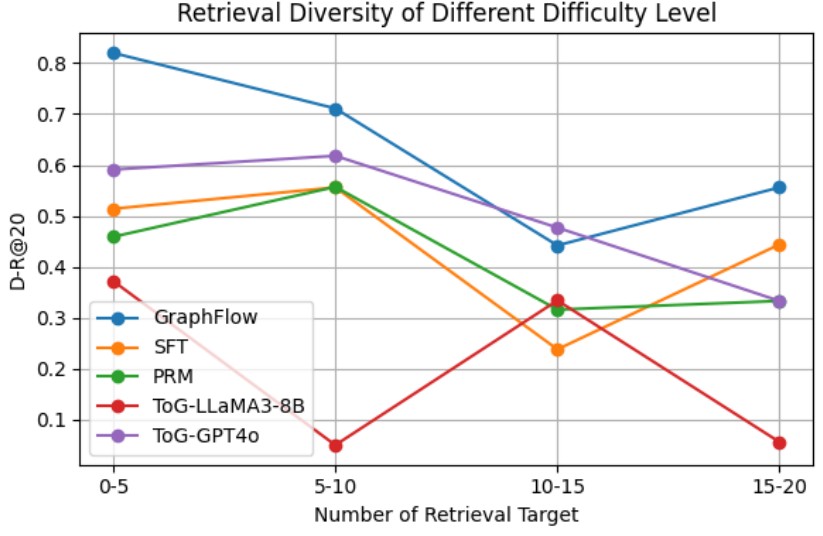

Figure 7: GraphFlow shows improved retrieval diversity on different difficulty levels of retrieval queries on STaRK-PRIME.

# E    More results of Hard Cases

We categorize the retrieval queries with different numbers of retrieval targets into 4 difficulty levels. We provide the performance of different KG-based RAG on STaRK-PRIME, STaRK-MAG, and STaRK-AMAZON at different difficulty levels. The results are shown in Figure 7, Figure 8, and Figure 9.

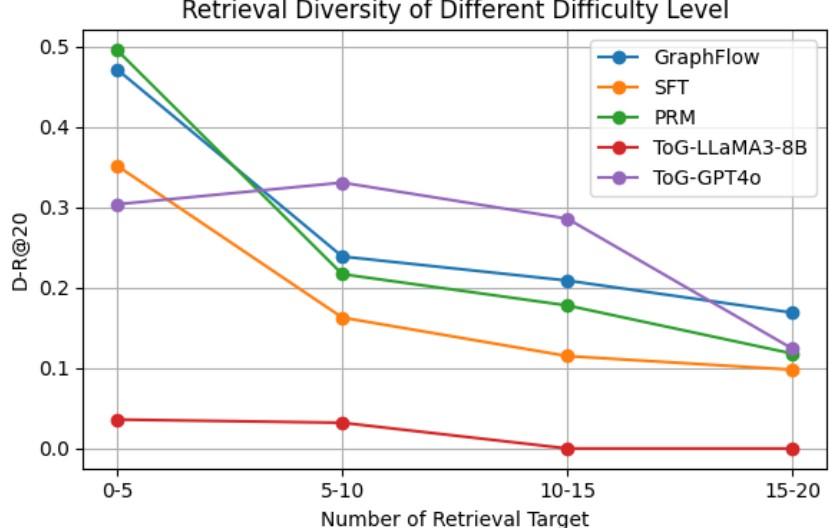

Figure 8: GraphFlow shows improved retrieval diversity on different difficulty levels of retrieval queries on STaRK-AMAZON.

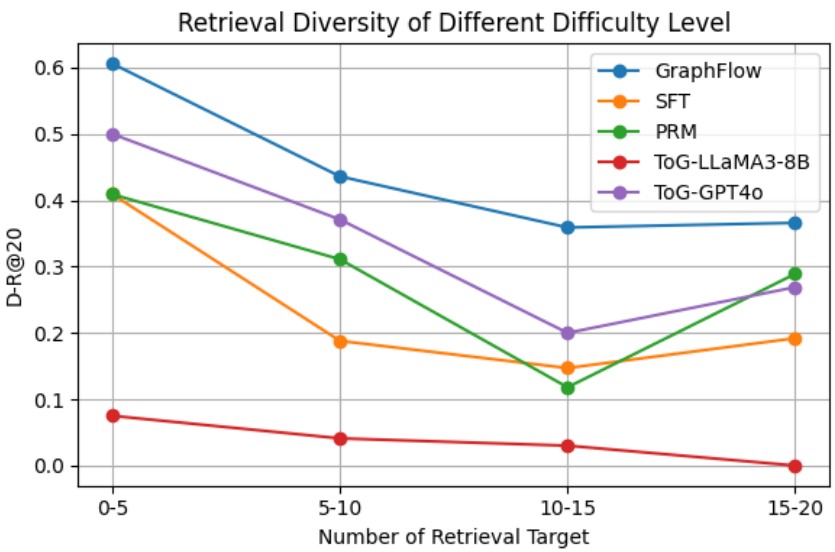

Figure 9: GraphFlow shows improved retrieval diversity on different difficulty levels of retrieval queries on STaRK-MAG.

## F Benchmark Information

We provide benchmark information in Table 5.

## G Computing Resources

We run all experiments on 8/16 NVIDIA-A800-SXM4-80GB GPUs and 56 Intel(R) Xeon(R) Platinum 8336C CPUs.

# H   Broader Impact and Limitations

GraphFlow introduces a novel framework for retrieval-augmented generation over text-rich knowledge graphs, enabling Large Language Models (LLMs) to reason more effectively through process supervision using GFlowNets. By modeling retrieval as a generative process that balances diverse and relevant paths, GraphFlow promotes both interpretability and coverage in knowledge-based reasoning. This has broad implications for applications such as scientific discovery, open-domain question answering, and medical decision support, where combining structured knowledge with free-text reasoning is crucial. Moreover, GraphFlow can serve as a foundation for future research in integrating generative decision-making with symbolic structures, thereby pushing forward the synergy between LLMs and knowledge graphs. One potential limitation is that we only evaluate the generalization ability of GraphFlow on two new domains.

