# OpenReview forum: "Can Knowledge-Graph-based Retrieval Augmented Generation Really Retrieve What You Need?"
_NeurIPS.cc/2025/Conference — NeurIPS 2025 spotlight_

### Official Review · Reviewer_yp42 · 2025-07-01

**Clarity:** 2
**Significance:** 2
**Originality:** 4
**Rating:** 5
**Confidence:** 5

**Summary:**

This paper proposed to use flow estimation as progress reward models in the training of KG-RAG LLMs.

**Questions:**

Questions:

Please refer to Weaknesses.

Also, since the training process relies on sampled trajectories who cover the final answer, does this means it eventually relies on terminal reward?
Considering recent progress of RL4LLM (e.g., the GRPO method) who only depends on terminal rewards , how different between intermediate supervised and terminal supervised in KG-RAG?

Suggestion:
Maybe finding a new way to model the flow estimation?

**Ethical Concerns:**

["NO or VERY MINOR ethics concerns only"]

**Final Justification:**

1. Well-written paper.
2. Interesting idea.
3. Novel and solid work.

The shortcoming of this paper, in my opinion, is its evaluation. It only evaluate on STaRK, which is a less common choice.
However, its goodness overcomes its shortness. I recommend acceptance.

**Limitations:**

yes

**Quality:**

2

**Strengths And Weaknesses:**

Strengths:
1. The idea is interesting: this paper propose to learn a flow function as progress reward models for KG-RAG. I like the idea of introducing intermediate rewards and the idea of using Gflow network/ flow matching.

----
Weaknesses:
1. **The learning method is problematic.** In the paper, the authors defined states to be a collection of documents. The flow estimation $F(s_{t+1})$ is calculated by putting all documents of $s_{t+1}$ into a LLM. More specifically, it is calculated by the last  token embedding transformed by a MLP. However, the conditional probability $P(s_{t+1}|s_{t})$ is calculated in a similar way. The only difference between these two is the MLP and the prompt. Within the same documents as input, I doubt how much difference we can have. The learning objective is the MSE between $\log F(s_{t})-\log F(s_{t+1})$ and $-\log P(s_{t+1}|s_{t})$. I don't think this makes sense.

2. **Weak Evaluation** The authors evaluated their method on the STaRK benchmark, which is a less common choice. Using STaRK has no problem, but I would like to see more results since I doubt the effectiveness of methodology. A more common choice of KG-RAG is either Freebase KG (datasets like WebQSP and CWQ), Wikidata KG (2WikiMultihopQA), or even Wikipedia (HotpotQA). The Wikipedia corpus has hyperlinks between entities and has a document for each entity, which exactly matches the description of text-attributed KG in the paper. Moreover, the baselines are weak. The ToG method is the strongest one among the baselines. However, according to my own experience, I cannot reproduce the result reported in the ToG paper. (Its official GitHub has many reproduction issues.) So, I doubt the effectiveness of this method.

---

> ### Author Rebuttal · Authors · 2025-07-31
>
> Thank you very much for your time and effort in providing valuable comments and suggestions! Our responses and clarification regarding the concerns are as follows:
>
> **Q1: The Learning method is problematic. The MSE-based learning objective in Eq. 7 does not make sense.**
>
> **A1**: Thank you for raising this concern. Our goal in GraphFlow is to learn a forward policy $P(s_{t+1}|s_{t})$ that generates diverse and high-quality retrieval results over text-rich KGs. To achieve this, we adopt an energy-based formulation (Eq. 5), which provides a principled way to assign scores to retrieval trajectories based on their rewards. To make this objective tractable, we leverage the detailed balance condition from GFlowNet theory [1,2], as shown in Eq. 6. We further introduce detailed balance with local exploration to ensure computational feasibility during optimization. This balance condition enforces consistency between the forward and backward policies and allows us to define a training signal over state transitions as shown in Eq. 7.
> Specifically, we:
>
> - Take the logarithm of both sides of Eq. 6. Assume a fixed backward policy $P_{B}(s_{t}|s_{t+1})=1$, which is valid under the irreversible trajectory construction used in our sampling process [3].
>
> - Expand the forward policy using the equation in Line 195, and **it finally leads to a mean squared error (MSE)-style objective (Eq. 7)**.
>
> - We also provide a more detailed background of Eq.5-Eq.7 in Section A and Section B of the Supplementary Material.
>
> **In summary, the MSE-like objective is a principled surrogate loss derived from our energy-based objective using well-established assumptions in the GFlowNet literature**.
>
> **Q2: The output between the flow estimation and policy network does not have difference since they receive the same document as input. The only difference between the flow estimation and policy network is the MLP and the prompt.**
>
> **A2**: We would like to clarify that the flow estimation $F(s_{t})$ and policy network $P(s_{t+1}|s_{t})$ receive different inputs and their output are also different. We define the state $s_{t}$ as the $t$-step information-seeking trajectory and the question. $F(s_{t})$ takes $s_{t}$ as input, while $P(s_{t+1}|s_{t})$ takes $s_{t}$ and candidate $a_{t}$ as input.
>
> Also, If their outputs were the same, then all candidate next states would have equal transition probabilities:$F(s_{t})=P(s^{1}_{t+1}|s_{t})=\cdots=P(s^{k}_{t+1}|s_{t})$. This would result in a random policy, and unable to distinguish between high-quality and low-quality candidates.
>
> **Thus, $F(s_{t})$ and $P(s_{t+1}|s_{t})$ have different inputs and outputs.** We have introduced the intuition and design of $F(s_{t})$ and $P(s_{t+1}|s_{t})$ in Section 3.2, Section 3.3 and Figure 2, with more details on the input, architecture, and prompts used for $F(s_{t})$ and $P(s_{t+1}|s_{t})$ provided in the Section B of Supplementary material.
>
> We hope our clarification could answer your question.
>
> **Q3: StaRK Dataset is not a common choice. More evaluations on Freebase KG (CWQ, WebQSP), and Wikidata KG (2WikiMultihopQA), or even Wikipedia (HotpotQA).**
>
> **A3**: STaRK is retrieval-centric and provides explicit ground-truth retrieval items for measuring retrieval accuracy and diversity, making it a good benchmark for our tasks (also pointed out by Reviewer jABU). The other mentioned datasets are Question-Answering (QA) datasets instead of retrieval-centric datasets:
>
> - **CWQ and WebQSP are not included** since: a). scenario misalignment between simple KG retrieval and text-rich KG retrieval, b). different settings between QA and retrieval, c). data quality issue due to incomplete KGs. **We refer reviewer yp42 to our response Q1 to reviewer PKMH for more detailed discussion and experiment analysis**.
>
> - **Wikidata KG (2WikiMultihopQA), or even Wikipedia (HotpotQA)** are not included since: a). 2WikiMultihopQA is also QA dataset and used by a recent arxiv paper [4] (uploaded after NeurIPS ddl). b). HotpotQA is also widely used for QA tasks. In the KG-based RAG domain, HotpotQA is usually used for KG construction [5,6,7], while we focus on the retrieval task.
>
> **Q4: The reproducible issue of ToG on CWQ and WebQSP.**
>
> **A4**: ToG employs CWQ and WebQSP for evaluation in their paper. But we reimplement it on the STaRK benchmark in our manuscript. **The performance divergence is due to the difference in the setting (simple KG v.s. text-rich KG)**. Additional evidence is that SubgraphRAG achieves very good performance on CWQ and WebQSP in their paper, but our reimplementation on STaRK shows that it lags behind baselines such as SFT, ToG, and PRM.
>
> **Q5: Since the training process relies on sampled trajectories who cover the final answer, does this means it eventually relies on terminal reward?**
>
> **A5**: We would like to clarify that the terminal reward in the training set is the one and only supervision signal for GraphFlow. Other baselines, such as SubgraphRAG, also use the final answer as supervision.
>
> **Q6: "Considering recent progress of RL4LLM (e.g., the GRPO method) who only depends on terminal rewards , how different between intermediate supervised and terminal supervised in KG-RAG?"**
>
> **A6**: **GraphFlow is quite different from GRPO as GraphFlow is off-policy while GRPO is on-policy**. But it is very interesting to explore GRPO in the KG-based RAG scenario. As for the difference between intermediate supervised and terminal supervised, our experiment shows that using step-wise supervision (SFT) lags behind GrahFlow (implicitly factorizes the terminal reward into step-wise supervision). With the GflowNet-based objective, GraphFlow can learn a policy that leads to diverse and accurate samples guided by factorized step-wise supervision.
>
> **Q7: Maybe find a new way to model the flow estimation?**
>
> **A7**: Thank you very much for the valuable questions. We welcome discussion about any concrete direction you might have in mind for exploring alternative flow estimation methods to enhance the KG-based RAG.
>
>
>
>
>
>
>
> [1]. GFlowNet Foundations, JMLR 2023.
>
> [2]. Trajectory balance: Improved credit assignment in GFlowNet. NeurIPS 2022.
>
> [3]. Amortizing intractable inference in large language model. ICLR 2024.
>
> [4]. KG-Infused RAG: Augmenting Corpus-Based RAG with External Knowledge Graphs. Arxiv: 2506.09542
>
> [5]. Knowledge Graph-Guided Retrieval Augmented Generation. Arxiv: 2502.06864.
>
> [6]. GFM-RAG: Graph Foundation Model for Retrieval Augmented Generation. Arxiv: 2502.01113
>
> [7]. KAG: Boosting LLMs in Professional Domains via Knowledge Augmented Generation. NAACL 2025.

---

> > ### Comment · Reviewer_yp42 · 2025-08-05
> > **Reply**
> >
> > I've raised my score.
> > I've re-read the paper and read the rebuttal.
> >
> > I found I have misunderstood Eq. (7). Now my concern is addressed.
> >
> > My suggestion is to add more benchmarks. Rebuttal A3 did not convince me. 2Wiki and HotpotQA are Open domain dataset, which means they can also evaluate retrieval performance, not limited in QA.
> >
> > But overall, it is a solid and novel work. I raised the score.

---

> > > ### Author Response · Authors · 2025-08-05
> > > **Excited to hear your concerns are addressed! Really appreciated that you could raise the score!**
> > >
> > > **We appreciate that you took the time to review our response and acknowledge that our work is solid and novel!** Thank you very much for your time and effort! We are excited that the misunderstanding and concerns have been addressed. In our future work, we will extend our framework to a more general setting, rather than a KG-based RAG.

---

### Official Review · Reviewer_jABU · 2025-07-02

**Clarity:** 3
**Significance:** 2
**Originality:** 3
**Rating:** 5
**Confidence:** 3

**Summary:**

The paper proposes GraphFlow, a new framework for improving Retrieval-Augmented Generation (RAG) using knowledge graphs (KGs). Existing KG-based RAG methods often fail to retrieve both accurate and diverse information for complex queries. Unlike prior approaches that require costly process-level supervision, GraphFlow uses a flow-based retrieval policy trained with only final outcome rewards. Inspired by GFlowNets, GraphFlow models retrieval as a multi-step decision process, jointly learning a policy and a flow estimator using a detailed balance objective. This enables effective exploration of high-reward regions in KGs. Experiments on the STaRK benchmark show that GraphFlow outperforms strong baselines (including GPT-4o) by around 10% in both accuracy and diversity, and generalizes well to unseen domains.

**Questions:**

- Why is GraphFlow deployed using LLaMA3-8B-Instruct while ToG uses LLaMA3-8B?
- I can't find the training/dev/test dataset statistics used for SFT, PRM and GraphFlow. Please make sure that this information is clearly provided in the final paper?

**Ethical Concerns:**

["NO or VERY MINOR ethics concerns only"]

**Final Justification:**

All my minor comments were addressed and I maintained my high score.

**Limitations:**

Yes

**Quality:**

3

**Strengths And Weaknesses:**

Strengths:
- The writing and presentation is clear.
- Their methodology is novel and well-motivated. Leveraging GFlowNets for this problem is an inspiring prospect.
- Their empirical results show strong performance over other supervised methods, making it a potentially practical system.

Weaknesses:
- Although STARK is a reasonably good benchmark for evaluating KG-based RAG systems, it is specifically design to test performance on a very high quality KG coupled with documents. The more general setting of KG-based RAG, in which a KG is automatically extracted from text [GraphRAG, HippoRAG, LightRAG, RAPTOR] and then leveraged for more effective retrieval, would be a more interesting and impactful testing ground for this framework. This is not grounds for rejection but a comment on the potential significance on this method.

---

> ### Author Rebuttal · Authors · 2025-07-31
>
> We really appreciate your time and effort in providing valuable comments. **We thank the reviewer for acknowledging our well-presented manuscript, well-motivated methodology, and strong empirical performance**. Our responses to your questions are summarized as follows:
>
> **Q1: [Furture work] Consider more general setting in KG-based RAG systerms where KGs are automatically extracted from raw texts.**
>
> **A1**: Thank you very much for this valuable suggestion. While our current work focuses on retrieval from text-rich, curated KGs, we agree that automatically extracting KGs from raw text is crucial for broader applicability. In future extensions of GraphFlow, we will explore two potential directions:
>
> - We will extend GraphFlow to access external tools, such as entity extraction and web search, to construct high-quality and text-rich KGs from web-sacle academic literature.
>
> - We plan to handle incomplete or noisy KGs by combining KG completion and retrieval. GraphFlow will not only traverse text-rich graphs but also refine entity connectivity and completeness.
>
> These extensions will enable GraphFlow to operate in more general real-world settings where high-quality KGs don't pre-exist.
>
>
> **Q2: Why is GraphFlow deployed using LLaMA3-8B-Instruct while ToG uses LLaMA3-8B?**
>
> **A2**: Thank you very much for pointing out this typo issue! We implement ToG using LLaMA3-8B-Instruct in our manuscript to keep a fair comparison. And we will correct this typo issue in the revised version.
>
> **Q3: Include the detailed training/dev/test dataset statistics used for SFT, PRM, and GraphFlow in the revised manuscript.**
>
> **A3**: Thank you very much for your valuable suggestions! We initially introduced the training/validation/test dataset split in Table 2 in the supplementary material. We will further include the detailed statistics of processed training/validation/test datasets to implement G-Retriever/SubgraphRAG/PRM/SFT/GraphFlow, the data format for model finetuning, and visualization of data instances in the Experiment section of the revised manuscript.
>
> We would like to thank you again for your comments and acknowledgement!

---

### Official Review · Reviewer_PkmH · 2025-07-03

**Clarity:** 2
**Significance:** 3
**Originality:** 3
**Rating:** 5
**Confidence:** 3

**Summary:**

The paper explores the use of Knowledge Graphs to enhance Retrieval-Augmented Generation for large language models. The authors propose a novel framework called GraphFlow that aims to efficiently retrieve accurate and diverse knowledge required for complex queries from text-rich KGs. GraphFlow employs a detailed balance objective with local exploration to jointly optimize a retrieval policy and a flow estimator. This approach allows GraphFlow to explore relevant regions of KGs, yielding diverse and accurate results. The framework is evaluated on the STaRK benchmark, demonstrating significant performance gains over existing KG-based RAG methods.

**Questions:**

1.	The supplementary material notes that only a handful of KG-based RAG models have been evaluated on the text-rich STaRK benchmark, whereas most prior work relies on simpler KGQA datasets such as CWQ and WebQSP. Could the authors explain why CWQ and WebQSP were omitted from the experiments? Reporting results on these widely used benchmarks would provide a clearer picture of the framework’s generalizability.
2.	Can the authors provide more examples of real-world applications where GraphFlow has been successfully implemented? This would help illustrate the practical benefits of the framework.

**Ethical Concerns:**

["NO or VERY MINOR ethics concerns only"]

**Final Justification:**

The rebuttal has addressed most of my concerns.

**Limitations:**

Paper discuss the limitations of GraphFlow on complexity and dependence on high-quality data.  Regarding potential negative societal impacts, the paper does not explicitly address this aspect.

**Paper Formatting Concerns:**

There are no formatting concerns in this paper.

**Quality:**

3

**Strengths And Weaknesses:**

Strengths:
+ The introduction of GraphFlow, which uses a detailed balance objective and local exploration, is a novel contribution that addresses the limitations of existing KG-based RAG methods. GraphFlow shows a 10% performance gain in both retrieval accuracy and diversity metrics compared to strong baselines, including GPT-4o.
+ The framework demonstrates strong generalization capabilities by effectively retrieving information from unseen KGs to support new-domain queries. The authors provide extensive experimental results on the STaRK benchmark, covering multiple domains such as e-commerce, academic, and biomedical KGs.

Weaknesses:
- The commonly used KGVQA datasets CWQ and WebQSP are not included in the experiments. Including results on these standard benchmarks would help better illustrate the generalizability and robustness of the proposed framework.
- The effectiveness of GraphFlow relies on the availability of high-quality, text-rich KGs, which may not always be accessible or well-maintained.

---

> ### Author Rebuttal · Authors · 2025-07-31
>
> We really appreciate your time and effort in providing valuable comments. We thank the reviewer for acknowledging the novelty of our method, extensive empirical evaluation, and significant performance gain. Our responses to your concerns are summarized as follows:
>
> **Q1: Could the authors explain why CWQ and WebQSP were omitted from the experiments?**
>
> **A1**: Thanks for the constructive suggestions! While CWQ and WebQSP are well-known KG‑based QA datasets, we did not include them in our current experiments due to the following reasons:
>
> - **Scenario Misalignment**: CWQ and WebQSP feature shallow, non-text-rich KGs, where entities are typically associated only with short phrases or labels. In contrast, our focus is on multi-hop retrieval from text-rich KGs, where entities connect to longer documents or passages. These two settings are fundamentally different (see Introduction, Lines 39–62).
>
> - **Different settings**: STaRK datasets are retrieval-centric and provide explicit ground-truth retrieval items for measuring retrieval accuracy and diversity. Meanwhile, CWQ and WebQSP are question-answering (QA) datasets. Since GraphFlow targets retrieval effectiveness, STaRK aligns more directly with our objective.
>
> - **Incomplete KGs in CWQ and WebQSP**: Although we could potentially treat the answer entities in CWQ and WebQSP as the ground-truth retrieval items, we find that many answer entities are missing in CWQ and WebQSP. **Given the dataset quality, it is very difficult to faithfully evaluate the retrieval performance of different KG-based RAG methods on CWQ and WebQSP**.  We show the statistics in Table 1.
>
> **Table 1: Dataset analysis of CWQ and WebQSP**
> | Dataset | Total Answers (test) | Missing Answers (test) | Missing Rate |
> |--------:|----------------------:|------------------------:|-------------:|
> | CWQ     |                6,676  |                 1,907   |       28.56% |
> | WebQSP  |               16,602  |                 7,094   |       42.73% |
>
> We find that SubgraphRAG [1] uses the shortest-path triplets and GPT4-annotated triplets as the ground-truth retrieval items. Although these ground-truths might not be accurate, we follow their settings and reimplement GraphFlow on WebQSP. The recall-rate are in Table 2 and the performances of baselines are quoted from SubgraphRAG.
>
> **Table 2: The retrieval performance on WebQSP**
>
> | Method          | Shortest Path Triplets | GPT‑4-annotated Triplets |
> |-----------------|------------------------|----------------|
> | Cosine similarity | 0.714               | 0.719          |
> | Retrieval‑rewrite‑answer | 0.058        | 0.062          |
> | RoG             | 0.713                 | 0.388          |
> | G‑Retriever     | 0.294                 | 0.325          |
> | GNN‑RAG         | 0.522                 | 0.405          |
> | SubgraphRAG     | 0.883                 | 0.865          |
> | PRM             | 0.841                 | 0.790          |
> | GraphFlow       | 0.923                 | 0.824          |
>
> Since retrieval from WebQSP mainly leverages structural information without rich textual information, it does not target the motivation of GraphFlow. However, our performance is still competitive, showing good generalization.
>
> **Q2: GraphFlow relies on high-quality and text-rich KGs, which are not easily accessible or well-maintained.**
>
> **A2**: Thank you for pointing this out.  We have illustrated the motivation of developing GraphFlow for diverse and accurate retrieval from text-rich KGs in the Introduction (Line 33-62). To further clarify, we will add more illustrative examples in the Introduction, where text-rich KGs are accessible and well-constructed. We will add a DeepResearch scenario for clarification, where LLMs need to identify relevant scientific literature to a scientific problem to write a report, draw inspiration, and propose a hypothesis. Here, the identification of relevant scientific literature involves retrieving diverse and accurate items from a paper-citation KG. In our future work, we will also explore the application of GraphFlow on the DeepResearch scenario.
>
> **Q3: Can the authors provide more examples of real-world applications that GraphFlow has been successfully implemented?**
>
> **A3**: Thanks for these constructive suggestions! In our manuscripts, we evaluate the proposed GraphFlow and other baselines on the STaRK benchmark, which consists of text-rich KGs from academic, e-commerce, and biology domains. Our experiments on STaRK have shown that the proposed GraphFlow can outperform the baselines by a large margin in terms of retrieval accuracy and diversity on text-rich KGs.
>
> We agree that exploring real-world applications of the GraphFlow framework is more interesting and promising, since the STaRK benchmark is still curated for research purposes. Though we haven’t yet deployed GraphFlow in these production environments due to time constraints during rebuttal, we believe integrating GraphFlow with AI-powered academic search is very interesting. A possible direction is to incorporate the GraphFlow framework with AI Scientist, which accelerates scientific innovation by retrieving diverse and relevant literature to yield inspiration for scientific problems.
>
>
> [1]. SIMPLE IS EFFECTIVE: THE ROLES OF GRAPHS AND LARGE LANGUAGE MODELS IN KNOWLEDGE-GRAPH-BASED RETRIEVAL-AUGMENTED GENERATION. ICLR 2025.

---

> > ### Comment · Reviewer_PkmH · 2025-08-05
> >
> > Thanks authors for their response. Most of my concerns have been adequately addressed in the rebuttal. I will raise my score accordingly.

---

> > > ### Author Response · Authors · 2025-08-06
> > > **Reply**
> > >
> > > Thank you very much for the response! We are glad that your concerns are addressed! We really appreciate your time and effort in reviewing our manuscript!

---

### Official Review · Reviewer_PQ1s · 2025-07-03

**Clarity:** 3
**Significance:** 2
**Originality:** 3
**Rating:** 5
**Confidence:** 4

**Summary:**

This paper tackles the problem of retrieval-augmented generation over text-enriched knowledge graphs for complex, multi-facet queries. The authors observe that prior KG-based RAG methods underperform when the query demands (i) fusing structural and textual evidence, and (ii) returning a diverse set of supporting items, and while procedural reward models could in principle give step-wise guidance, they need costly process-level supervision unavailable for large KGs Thus, the paper introduces GraphFlow, a GFlowNet-inspired framework that models KG retrieval as a stochastic decision process. GraphFlow jointly learns a retrieval policy that samples multi-hop sub-graphs, and a flow estimator that factorizes the terminal reward backward onto each intermediate step, promoting both accuracy and diversity without explicit step-wise labels. Experiments on the STaRK benchmark show that GraphFlow exceeds strong baselines, becoming an effective and robust solution for complex KG-based RAG scenarios.

**Questions:**

Why can this method outperform ToG by such a large margin? Is it because of the gap between the limited context window size and the large degree of nodes in STaRK, or other reasons?

**Ethical Concerns:**

["NO or VERY MINOR ethics concerns only"]

**Final Justification:**

The authors provided sufficient experiments to address my concerns, covering different aspects including baseline comparison, in-depth analysis, and algorithm efficiency. The results make a good supplement to the original paper, demonstrating its solidity and completeness. Overall, I recommend accepting this paper due to the following highlights:
1. The method efficiently and effectively addresses the diversity problem in complex retrieval, which is an important aspect often overlooked by similar works.
2. The algorithm is solid and interpretable. During the rebuttal, the authors add experiments to quantify the usefulness of retrieved documents in the middle steps. This strengthens their claim in the title, adding good supportive evidence apart from end-to-end blackbox QA performance.
3. The related works sorted in this paper are very clear and helpful, which would make this paper a good reference to guide future researchers.
4. The added experiments and explanations cover a complete aspects of a classical good paper, including ablations, interpretation, and efficiency analysis. The completeness and efficiency of the method convince me that it is easy and safe to use for this task.

**Limitations:**

yes

**Quality:**

2

**Strengths And Weaknesses:**

Strengths:

- This paper is very well written and joyful to read. It clearly and reasonably frames related works to retrieval-based and agent-based and uses proper examples to illustrate their idea, showing the authors' good understanding of this area.

- The paper's core technical idea for providing step-wise guidance using a trained LLM is both intuitive and elegant. The experiments also demonstrate the effectiveness and generalizability of the proposed method.

Weaknesses:

- Missing baselines and analysis. The experimental section omits some widely‐used STaRK baselines such as AVATAR [1], MFAR [2], KAR [3], and MOR [4]. My own inspection shows that on the Amazon and MAG domains the proposed method still lags these baselines by a noticeable margin. The impressive result of this paper lies in the PRIME domain, where by far it's the best results I've seen and leads by a large margin. However, there is a lack of analysis of this domain-variant result in this paper.

- Does the paper sufficiently answer the question in the title? More analysis is needed to illustrate whether a retrieval method (previous and proposed) retrieves useful documents. For example, you may use SePer [5] to quantitatively estimate the utility of information at the intermediate steps.

- Efficiency comparison unclear. One of the key claimed advantages is the efficiency of using the flow estimator for graph retrieval. However, there is a lack of experiments on the cost in time and money compared to retrieval-based and other agent-based retrieval methods.

Would consider improving the score if more sufficient experiments are shown.

[1]  Avatar: Optimizing llm agents for tool usage via contrastive reasoning

[2]  Multi-field adaptive retrieval

[3] Knowledge-Aware Query Expansion with Large Language Models for Textual and Relational Retrieval

[4] Mixture of Structural-and-Textual Retrieval over Text-rich Graph Knowledge Bases

[5] SePer: Measure Retrieval Utility Through The Lens Of Semantic Perplexity Reduction

---

> ### Author Rebuttal · Authors · 2025-07-31
>
> We really appreciate your time and effort in reviewing and providing valuable comments for us to improve our manuscript. Especially, we like the recent $\Delta_{seper}$ metric introduced by the reviewer. We will add the $\Delta_{seper}$ results to Experiment Section and other analysis in their corresponding sections. Our responses are as follows:
>
> **Q1: More comparison with the baselines and analysis of the domain-variant result in the paper.**
>
> **A1**: We will add a section in the revised manuscript to compare GraphFlow with recent STaRK baselines, including AVATAR, MFAR, KAR, and MoR, and compare their empirical performance. We summarize the main difference below:
>
> - **Comparison with AVATAR [1]**: AVATAR is an agentic framework built on API-based LLMs (e.g., GPT‑4, Claude‑3‑Opus) and leverages external tools like WEB_SEARCH and WIKI_SEARCH. It also employ an API-model-based Comparator module to optimize retrieval instructions. Its implementation incurs substantial token and computation costs (e.g., very high API usage), which we confirm is a practical barrier to reproduction (in Appendix D, p. 20 of the AVATAR paper). In contrast, GraphFlow relies on fine-tuning an 8B-scale LLM to traverse text-rich KGs directly, avoiding external APIs or tool dependencies. **This leads to significantly lower resource consumption while maintaining comparable performance to the GPT‑4–based AVATAR baseline** (Table 4 in AVATAR paper).
>
> - **Comparison with MFAR/KAR/MoR [2,3,4]**: They integrate multiple retrievers while GraphFlow only employs one retriever.  MFAR ensembles multiple retrieval models (lexical and dense), with learned weighting functions. KAR and MoR integrate both text and graph retrievers in intermediate retrieval steps. Differently, GraphFlow relies on a pure graph retriever to retrieve from text-rich KGs. While integration of multiple retrievers leads to promising results, **GraphFlow still shows strong performance over other one-retriever baselines**. Given time and complexity constraints, we leave “GraphFlow + multiple retrievers” to future work.
>
> - **Domain-Variant Analysis**: PRIME SKB is significantly denser (avg. degree=125.2) with more relation types and textual annotations, compared to MAG (avg. degree=43.5) and Amazon (avg. degree=18.2). GraphFlow’s graph-based inductive bias aligns closely with PRIME’s dense structure, making it particularly effective in that domain. Performance gaps on MAG and Amazon likely stem from their sparser graph structure. This also explains why other agent-based baselines show similar trends. This finding also motivates us to explore “GraphFlow + multiple retrievers” in future work.
>
> **Q2: More analysis on the retrieval steps. For example, using SePer to evaluate the utility of information at the intermediate steps**
>
> **A2**: Thank you for this insightful suggestion! In the current manuscript, we evaluate retrieval performance using conventional metrics like Hit@1 and Hit@5. Our results highlight that both retriever-based and agent-based baselines struggle to retrieve diverse and accurate contents from text-rich KGs, leading to our motivation for proposing GraphFlow. We appreciate the reviewer introducing **SePer**, a recent ICLR 2025 metric [5] for evaluating retrieval utility by measuring semantic perplexity reduction after retrieval: $\Delta_{seper}=P_{M}(a|q,D)-P_{M}(a|q)$. where 𝑀 is an LLM (here we use LLaMA3–8B–Instruct),  𝑞 is the question, 𝐷 is retrieved content, and 𝑎 is the ground truth answer.
>
> In STaRK benchmark, since there is no ground-truth answer, we use the title or summarized description of the ground-truth retrieval item as 𝑎. We design the following metrics for comprehensive evaluation, and we report their mean and std.
> - $Step \Delta_{seper}$: the $\Delta_{seper}$ score per retrieval step,
> - $Answer \Delta_{seper}$: the $\Delta_{seper}$ score of the final retrieval results.
>
> As shown in Table 1-3 below, GraphFlow consistently achieves higher $Step \Delta_{seper}$ and $Answer \Delta_{seper}$ than baselines, demonstrating stronger information utility during retrieval. This confirms our motivation:
> - Retrieving from text-rich KGs indeed poses huge challenges to current KG-based RAG methods.
> - GraphFlow can significantly improve the information utility when retrieving from text-rich KGs.
>
> **Table 1: $\Delta_{seper}$ on Prime.**
> | Method               | $Step \Delta_{seper}$      | $Answer \Delta_{seper}$      |
> |---------------------------|----------------------|------------------------|
> | ToG + GPT‑4              | 0.065 ± 0.125        | 0.148 ± 0.165          |
> | ToG + LLaMA3‑8B‑Instruct  | 0.009 ± 0.102        | 0.021 ± 0.106          |
> | SFT                       | 0.062 ± 0.132        | 0.158 ± 0.183          |
> | PRM                       | 0.057 ± 0.106        | 0.131 ± 0.174          |
> | G‑Retriever               | —                    | 0.029 ± 0.117          |
> | SubgraphRAG               | —                   | 0.046 ± 0.083          |
> | GraphFlow                 | 0.091 ± 0.147        | 0.206 ± 0.192          |
>
> **Table 2: $\Delta_{seper}$ on Amazon.**
> | Method                      | $Step \Delta_{seper}$       | $Answer \Delta_{seper}$       |
> |-----------------------------|------------------------|-------------------------|
> | ToG + GPT‑4                 | 0.031 ± 0.109          | 0.092 ± 0.128           |
> | ToG + LLaMA3‑8B‑Instruct     | 0.010 ± 0.118          | 0.046 ± 0.149           |
> | SFT                         | 0.079 ± 0.151          | 0.141 ± 0.160           |
> | PRM                         | 0.029 ± 0.089          | 0.071 ± 0.112           |
> | G‑Retriever                 | —                      | 0.024 ± 0.110           |
> | SubgraphRAG                 | —                      | 0.021 ± 0.093           |
> | GraphFlow           | 0.097 ± 0.158          | 0.219 ± 0.257           |
>
> **Table 3: $\Delta_{seper}$ on MAG.**
> | Method                      | $Step \Delta_{seper}$       | $Answer \Delta_{seper}$      |
> |-----------------------------|------------------------|-------------------------|
> | ToG + GPT‑4                 | 0.041 ± 0.172          | 0.065 ± 0.150           |
> | ToG + LLaMA3‑8B‑Instruct     | 0.068 ± 0.108          | 0.105 ± 0.146           |
> | SFT                         | 0.035 ± 0.101          | 0.084 ± 0.095           |
> | PRM                         | 0.037 ± 0.117          | 0.060 ± 0.115           |
> | G‑Retriever                 | —                      | 0.012 ± 0.089           |
> | SubgraphRAG                 | —                      | 0.039 ± 0.076           |
> | GraphFlow                   | 0.081 ± 0.137          | 0.145 ± 0.112           |
>
> Notice that all the methods have high variance in $\Delta_{seper}$ scores. This may be due to
>
> - High variance in natural language entailment when calculating  $\Delta_{seper}$ scores;
> - For all methods, we observe that some retrieval samples have negative $\Delta_{seper}$, indicating a negative impact on question answering when retrieving bad contents.
>
> **Q3: Efficiency comparison**.
>
> **A3**: When we say “to further enhance training efficiency” (Line 85), we refer to making the **training process** of GraphFlow both scalable and GPU-friendly. Applying the GFlowNet objective directly to large KGs introduces prohibitive computation and often triggers out-of-memory errors (Section 3.2, Line 178). To address the efficiency issue, we employ detailed balance with local exploration strategy to make the training process of GraphFlow efficient and scalable.  **We also report retrieval latency and inference cost at test time in Table 4**. SubgraphRAG enjoys fast inference speed since it employs a light-weight retriever. However, its retrieval performance on text-rich KGs is limited as reported in Table 1 in our manuscript.
>
> Table 4: Inference-time of different methods.
> | Method        | Inference Time |
> |---------------|----------------|
> | SubgraphRAG   | 0.094 s        |
> | G-Retriever   | 6.79 s        |
> | ToG + LLaMA   | 10.35 s        |
> | ToG + GPT‑4   | 14.49 s        |
> | PRM           | 8.94 s         |
> | SFT           | 4.56 s         |
> | GraphFlow     | 4.73 s         |
>
> **Q4: Is the performance gain of GraphFlow over ToG due to the document cutoff?**
>
> **A4**: When implemented with LLaMA3-8B-Instruct, ToG shares the same context length limit as GraphFlow. Moreover, we cut off the entity document to a maximum of 500 tokens when implementing ToG, SFT, PRM, and GraphFlow, to ensure implementation feasibility and a fair comparison. Thus, we could say that the performance gain of GraphFlow over ToG is not due to the document cutoff. We will add this analysis to our revised manuscripts.
>
> [1]. AvaTaR: Optimizing LLM Agents for Tool Usage via Contrastive Reasoning. NeurIPS 2024.
>
> [2]. MFAR: Multi-field adaptive retrieval. ICLR 2025.
>
> [3]. KAR: Knowledge-Aware Query Expansion with Large Language Models for Textual and Relational Retrieval. NAACL 2025.
>
> [4]. MoR: Mixture of Structural-and-Textual Retrieval over Text-rich Graph Knowledge Bases. NAACL 2025.
>
> [5]. SePer: Measure Retrieval Utility Through The Lens Of Semantic Perplexity Reduction. ICLR 2025.

---

> > ### Comment · Reviewer_PQ1s · 2025-08-01
> >
> > Thanks for the extensive experiments and justifications. They effectively address my concerns. Will improve my rating.

---

> > > ### Author Response · Authors · 2025-08-01
> > > **Excited to hear that your concerns are addressed!**
> > >
> > > We are excited to hear that our analysis has effectively addressed your concerns! We appreciate your constructive suggestions and comments, which help us improve the manuscript.

---

### Comment · Area_Chair_FC1T · 2025-08-04

Dear reviewers,

When you have a moment, please take a look at the authors’ rebuttal and update your final scores.

Best wishes, AC

---

### Note · Authors · 2025-08-13

We sincerely thank the SACs, ACs, and Reviewers for their time and effort in reviewing our work and engaging in the rebuttal process. In particular, we deeply appreciate the Reviewers for their insightful comments, valuable suggestions, and active interactions during the rebuttal, which significantly helped us improve the manuscript.

We have revised the manuscript from two main perspectives:

1. Enhanced empirical evaluation. We added the Seper score to quantify the information utility of our methods during retrieval (Reviewer PQ1s), reported retrieval results on WebQSP (Reviewer PKmh), and included an inference-time efficiency comparison (Reviewer PQ1s).

2. Improved clarification. We incorporated additional comparisons with related work (Reviewer PQ1s), domain-specific analysis (Reviewer PQ1s), benchmark analysis (Reviewers PkmH and jABU), future work discussion (Reviewers jABU and PkmH), real-world application scenarios (Reviewer PkmH), and further details on evaluation and methodology (Reviewer yp42).

During the rebuttal, Reviewers PQ1s and yp42 updated their evaluations, and we are grateful that all Reviewers ultimately provided positive assessments of our work.

---

### Decision · Program_Chairs · 2025-09-17

**Decision:**

Accept (spotlight)

**Comment:**

This paper introduces GraphFlow, a novel framework for enhancing knowledge-graph-based retrieval-augmented generation (KG-RAG). The work makes a technically solid and empirically validated contribution, offering a creative and interpretable methodology that addresses a recognized challenge in KG-based retrieval.

Reviewers highlight the paper’s strength in novelty and clarity, and commend the authors for the extensive clarifications and new experiments provided during rebuttal, which effectively resolved concerns regarding baselines, retrieval utility, and efficiency.

That said, the evaluation remains somewhat narrow in scope, and evidence of practical deployment on noisier, real-world KGs is still limited. With broader cross-dataset validation and demonstrations in more challenging settings, this line of work could reach an even higher level of impact.

Based on the overall assessment, a spotlight acceptance is recommended.